# ONE-PROMPT-ONE-STORY: FREE-LUNCH CONSISTENT TEXT-TO-IMAGE GENERATION USING A SINGLE PROMPT

**Tao Liu[1], Kai Wang[2]\*, Senmao Li[1], Joost van de Weijer[2], Fahad Shahbaz Khan[3,4]**
**Shiqi Yang[5], Yaxing Wang[6], Jian Yang[1], Ming-Ming Cheng[1]**
[1]VCIP, CS, Nankai University,  [2]Computer Vision Center, Universitat Autònoma de Barcelona
[3]Mohamed bin Zayed University of AI,  [4]Linkoping University,  [5]SB Intuitions, SoftBank
[6]Nankai International Advanced Research Institute (Shenzhen Futian), Nankai University
`{ltolcy0, senmaonk, shiqi.yang147.jp}@gmail.com, {kwang, joost}@cvc.uab.es`
`fahad.khan@liu.se, {yaxing, csjyang, cmm}@nankai.edu.cn`

## ABSTRACT

Text-to-image generation models can create high-quality images from input prompts. However, they struggle to support the consistent generation of identity-preserving requirements for storytelling. Existing approaches to this problem typically require extensive training in large datasets or additional modifications to the original model architectures. This limits their applicability across different domains and diverse diffusion model configurations. In this paper, we first observe the inherent capability of language models, coined *context consistency*, to comprehend identity through context with a single prompt. Drawing inspiration from the inherent *context consistency*, we propose a novel *training-free* method for consistent text-to-image (T2I) generation, termed "One-Prompt-One-Story" (*1Prompt1Story*). Our approach *1Prompt1Story* concatenates all prompts into a single input for T2I diffusion models, initially preserving character identities. We then refine the generation process using two novel techniques: *Singular-Value Reweighting* and *Identity-Preserving Cross-Attention*, ensuring better alignment with the input description for each frame. In our experiments, we compare our method against various existing consistent T2I generation approaches to demonstrate its effectiveness through quantitative metrics and qualitative assessments. Code is available at https://github.com/byliutao/1Prompt1Story.

## 1 INTRODUCTION

Text-based image generation (T2I) (Ramesh et al., 2022; Saharia et al., 2022; Rombach et al., 2022; Ma et al., 2024; Su et al., 2023) aims to generate high-quality images from textual prompts, depicting various subjects in various scenes. The ability of T2I diffusion models to maintain *subject consistency* across a wide range of scenes is crucial for applications such as animation (Hu, 2024; Guo et al., 2024), storytelling (Yang et al., 2024; Gong et al., 2023; Cheng et al., 2024), video generation models (Khachatryan et al., 2023; Blattmann et al., 2023) and other narrative-driven visual applications. However, achieving consistent T2I generation remains a challenge for existing models, as shown in Fig. 1 (up).

Recent studies tackle the challenge of maintaining subject consistency through diverse approaches. Most methods require time-consuming training on large datasets for clustering identities (Avrahami et al., 2023), learning large mapping encoders (Gal et al., 2023b; Ruiz et al., 2024), or performing fine-tuning (Ryu, 2023; Kopiczko et al., 2024), which carries the risk of inducing language drift (Heng & Soh, 2024; Wu et al., 2024a; Huang et al., 2024), etc. Several recent training-free approaches (Tewel et al., 2024; Zhou et al., 2024) demonstrate remarkable results in generating images with consistent subjects by leveraging shared internal activations from the pre-trained models. These methods require extensive memory resources or complex module designs to strengthen

---

\*: Corresponding authors.

"A hyper-realistic digital painting of a fairy[0], dressed in a cloak of spider silk[1], wearing a garland of fireflies[2]."

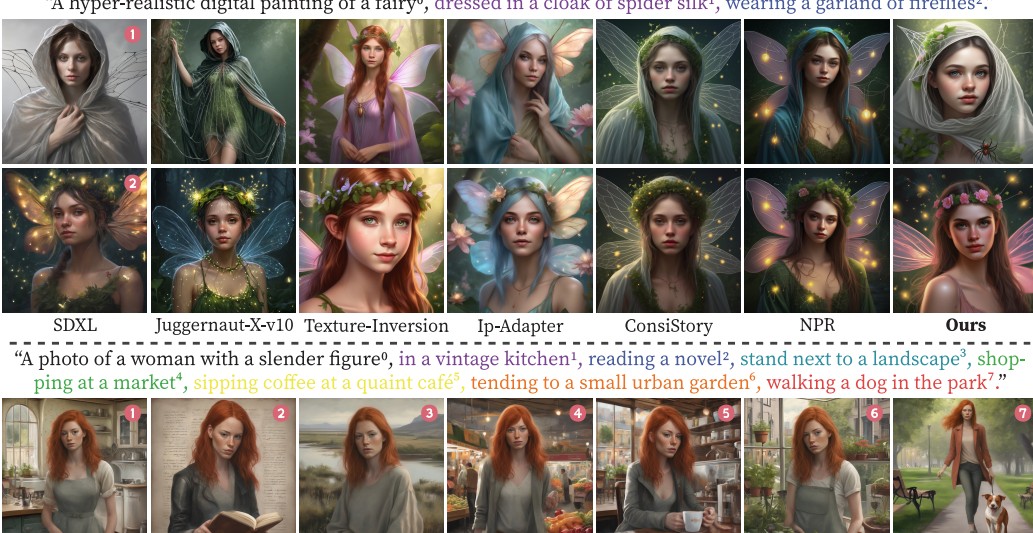

SDXL   Juggernaut-X-v10   Texture-Inversion   Ip-Adapter   ConsiStory   NPR   **Ours**

"A photo of a woman with a slender figure[0], in a vintage kitchen[1], reading a novel[2], stand next to a landscape[3], shopping at a market[4], sipping coffee at a quaint café[5], tending to a small urban garden[6], walking a dog in the park[7]."

Figure 1: **Existing methods (up)** encounter challenges in consistent T2I generation. T2I models such as SDXL (Podell et al., 2023) and Juggernaut-X-v10 (RunDiffusion, 2024) often exhibit noticeable identity *inconsistency* across generated images. Although recent methods including IP-Adapter and ConsiStory have improved *identity consistency*, they lost the alignment between the generated images and corresponding input prompts. **Additional results of our *1Prompt1Story* (down)** demonstrate superior consistency without compromising the alignment between text and images.

the T2I diffusion model to generate satisfactory consistent images. However, they all neglect the inherent property of long prompts that identity information is implicitly maintained by context understanding, which we refer to as the *context consistency* of language models. For example, the dog object in "A dog is watching the movie. Afterward, the dog is lying in the garden." can be easily understood as the same without any confusion since it appears in the same paragraph and is connected by the context. We take advantage of this inherent feature to eliminate the requirement of additional finetuning or complicated module design.

Observing the inherent *context consistency* of language models, we propose a novel approach to generate images with consistent characters using a single prompt, termed *One-Prompt-One-Story* (*1Prompt1Story*). Specifically, *1Prompt1Story* consolidates all desired prompts into a single longer sentence, which starts with an *identity prompt* that describes the corresponding identity attributes and continues with subsequent *frame prompts* describing the desired scenarios in each frame. We denote this first step as *prompts consolidation*. By reweighting the consolidated prompt embeddings, we can easily implement a basic method *Naive Prompt Reweighting* to adjust the T2I generation performance, and this approach inherently achieves excellent identity consistency. Fig. 1 (up, the 6th column) illustrates two examples, each featuring an image generated with different frame descriptions within a single prompt by reweighting the frame prompt embeddings. These examples demonstrate that *Naive Prompt Reweighting* is able to maintain identity consistency with various scenario prompts. However, this basic approach does not guarantee strong text-image alignment for each frame, as the semantics of each frame prompt are usually intertwined within the consolidated prompt embedding (Radford et al., 2021). To further enhance text-image alignment and identity consistency of the T2I generative models, we introduce two additional techniques: *Singular-Value Reweighting* (SVR) and *Identity-Preserving Cross-Attention* (IPCA). The *Singular-Value Reweighting* aims to refine the expression of the prompt of the current frame while attenuating the information from the other frames. Meanwhile, the strategy *Identity-Preserving Cross-Attention* strengthens the consistency of the subject in the cross-attention layers. By applying our proposed techniques, *1Prompt1Story* achieves more consistent T2I generation results compared to existing approaches.

In the experiments, we extend an existing consistent T2I generation benchmark as *ConsiStory+* and compare it with several state-of-the-art methods, including ConsiStory (Tewel et al., 2024), Story-Diffusion (Zhou et al., 2024), IP-Adapter (Ye et al., 2023), etc. Both qualitative and quantitative

performance demonstrate the effectiveness of our method *1Prompt1Story*. In summary, the main contributions of this paper are:

- To the best of our knowledge, we are the first to analyze the overlooked ability of language models to maintain inherent *context consistency*, where multiple frame descriptions within a single prompt inherently refer to the same subject identity.

- Based on the *context consistency* property, we propose *One-Prompt-One-Story* as a novel *training-free* method for consistent T2I generation. More specifically, we further propose *Singular-Value Reweighting* and *Identity-Preserving Cross-Attention* techniques to improve text-image alignment and subject consistency, allowing each frame prompt to be individually expressed within a single prompt while maintaining a consistent identity along with the *identity prompt*.

- Through extensive comparisons with existing consistent T2I generation approaches, we confirm the effectiveness of *1Prompt1Story* in generating images that consistently maintain identity throughout a lengthy narrative over our extended *ConsiStory+* benchmark.

## 2 RELATED WORK

**T2I personalized generation.** T2I personalization is also referred to *T2I model adaptation*. This aims to adapt a given model to a *new concept* by providing a few images and binding the new concept to a unique token. As a result, the adaptation model can generate various renditions of the new concept. One of the most representative methods is DreamBooth (Ruiz et al., 2023), where the pre-trained T2I model learns to bind a modified unique identifier to a specific subject given a few images, while it also updates the T2I model parameters. Recent approaches (Kumari et al., 2023; Han et al., 2023b; Shi et al., 2023) follow this pipeline and further improve the quality of the generation. Another representative, Textual Inversion (Gal et al., 2023a), focuses on learning new concept tokens instead of fine-tuning the T2I generative models. Textual Inversion finds new pseudo-words by conducting personalization in the text embedding space. The coming works (Dong et al., 2022; Voynov et al., 2023; Han et al., 2023a; Zeng et al., 2024) follow similar techniques.

**Consistent T2I generation.** Despite recent advances, T2I personalization methods often require extensive training to effectively learn modifier tokens. This training process can be time-consuming, which limits their practical impact. More recently, there has been a shift towards developing consistent T2I generation approaches (Wang et al., 2024b;a), which can be considered a specialized form of T2I personalization. These methods mainly focus on generating human faces that possess semantically similar attributes to the input images. Importantly, they aim to achieve this identity-preserving T2I generation without the need for additional fine-tuning. They mainly take advantage of PEFT techniques (Ryu, 2023; Kopiczko et al., 2024) or pre-training with large datasets (Ruiz et al., 2024; Xiao et al., 2023) to learn the image encoder to be customized in the semantic space. For example, PhotoMaker (Li et al., 2023b) enhances its ability to extract identity embeddings by fine-tuning part of the transformer layers in the image encoder and merging the class and image embeddings. The Chosen One (Avrahami et al., 2023) utilizes an identity clustering method to iteratively identify images with a similar appearance from a set of images generated by identical prompts.

However, most consistent T2I generation methods (Akdemir & Yanardag, 2024; Wang et al., 2024a) still require training the parameters of the T2I models, sacrificing compatibility with existing pre-trained community models, or fail to ensure high face fidelity. Additionally, as most of these systems (Li et al., 2023b; Gal et al., 2023b; Ruiz et al., 2024) are designed specifically for human faces, they encounter limitations when applied to non-human subjects. Even for the state-of-the-art approaches, including StoryDiffusion (Zhou et al., 2024), The Chosen One (Avrahami et al., 2023) and ConsiStory (Tewel et al., 2024), they either require time-consuming iterative clustering or high memory demand in generation to achieve identity consistency.

**Storytelling.** Story generation (Li et al., 2019; Maharana et al., 2021), also referred to as storytelling, is one of the active research directions that is highly related to character consistency. Recent researches (Tao et al., 2024; Wang et al., 2023) have integrated the prominent pre-trained T2I diffusion models (Rombach et al., 2022; Ramesh et al., 2022) and the majority of these approaches require intense training over storytelling datasets. For example, Make-a-Story (Rahman et al., 2023) introduces a visual memory module designed to capture and leverage contextual in-

formation throughout the storytelling process. StoryDALL-E (Maharana et al., 2022) extends the story generation paradigm to story continuation, using DALL-E capabilities to achieve substantial improvements over previous GAN-based methodologies. Note that the story continuation shares similarities with consistent Text-to-Image generation by using reference images. However, current consistent T2I generation methods prioritize preserving human face identities, whereas story continuation involves supporting various subjects or even multiple subjects within the generated images.

In this paper, our proposed consistent T2I framework, *1Prompt1Story*, diverges significantly from previous approaches in storytelling and consistent T2I generation methods. We explore the inherent *context consistency* property in language models instead of finetuning large models or designing complex modules. Importantly, it is compatible with various T2I generative models, since the properties of the text model are independent of the specific generation model used as the backbone.

## 3 METHOD

Consistent T2I generation aims to generate a set of images depicting consistent subjects in different scenarios using a set of prompts. These prompts start with an *identity prompt*, followed by the *frame prompts* for each subsequent visualization frame. In this section, we first empirically show that different frame descriptions included in a concatenated prompt can maintain identity consistency due to the inherent *context consistency* property of language models. We examine this observation through comprehensive analyses in Sec. 3.1 and propose the basic *Naive Prompt Reweighting* pipeline of our method *1Prompt1Story*. Following that, to ensure that each frame description within the prompt is expressed individually while diminishing the impact of other *frame prompts*, we introduce *Singular-Value Reweighting* and *Identity-Preserving Cross-Attention* in Sec. 3.2. The illustration of *1Prompt1Story* is shown in Fig. 4 and Algorithm 1 in the Appendix.

### 3.1 CONTEXT CONSISTENCY

**Latent Diffusion Models.** We build our approach on the SDXL (Podell et al., 2023) model, a latent diffusion model that contains two main components: an autoencoder (i.e., an encoder $\mathcal{E}$ and a decoder $\mathcal{D}$ ) and a diffusion model (i.e., $\epsilon_\theta$ parameterized by $\theta$). The model $\epsilon_\theta$ is trained with the following loss function:

$$L_{LDM} := \mathbb{E}_{z_0 \sim \mathcal{E}(x), \epsilon \sim \mathcal{N}(0,1), t \sim \text{Uniform}(1,T)} \left[ \|\epsilon - \epsilon_\theta(z_t, t, \tau_\xi(\mathcal{P}))\|_2^2 \right], \tag{1}$$

where $\epsilon_\theta$ is a UNet that conditions on the latent variable $z_t$, a timestep $t \sim \text{Uniform}(1, T)$, and a text embedding $\tau_\xi(\mathcal{P})$. In text-guided diffusion models, images are generated based on a textual condition, with $\mathcal{C} = \tau_\xi(\mathcal{P}) \in \mathbb{R}^{M \times D}$, where $M$ is the number of tokens, $D$ is the feature dimension of each token, and $\tau_\xi$ is the CLIP text encoder (Radford et al., 2021)[1]. For a given input, the model $\epsilon_\theta(z_t, t, \mathcal{C})$ produces a cross-attention map. Let $f_{z_t}$ denote the feature map output from $\epsilon_\theta$. We can obtain a query matrix $Q = l_Q(f_{z_t})$ using the projection network $l_Q$. Similarly, the key matrix $\mathcal{K}$ is computed from the text embedding $\mathcal{C}$ using another projection network $l_\mathcal{K}$, such that $\mathcal{K} = l_\mathcal{K}(\mathcal{C})$. The cross-attention map $\mathcal{A}_t$ is then calculated as: $\mathcal{A}_t = softmax(Q \cdot \mathcal{K}^T / \sqrt{d})$, where $d$ is the dimension of the query and key matrices. The entry $[\mathcal{A}_t]_{ij}$ represents the attention weight of the $j$-th token to the $i$-th token.

**Problem Setups.** In the T2I diffusion models, the text embedding $\mathcal{C} = \tau_\xi(\mathcal{P}) \in \mathbb{R}^{M \times D}$ is with $M$ tokens. The $M$ tokens contain a start token [SOT] , followed by $|\mathcal{P}|$ tokens corresponding to the prompt, and $M - |\mathcal{P}| - 1$ padding end tokens [EOT] . Previous consistent T2I generation works (Avrahami et al., 2023; Tewel et al., 2024; Zhou et al., 2024) generate images from a set of $N$ prompts. This set of prompts starts with an *identity prompt* $\mathcal{P}_0$ that describes the relevant attribute of the subject and continues with multiple frame prompt $\mathcal{P}_i$, where $i = 1, \ldots, N$ describes each frame scenario. However, this separate generation pipeline ignores the inherent language property, i.e., the *context consistency*, by which identity is consistently ensured by the context information inherent in language models. This property stems from the self-attention mechanism within Transformer-based text encoders (Radford et al., 2021; Vaswani et al., 2017), which allows learning the interaction between phrases in the text embedding space.

---

[1]SDXL uses two text encoders and concatenate the embeddings as the final input. $M = 77$ by default.

In the following, we analyze the *context consistency* under different prompt configurations in both textual space and image space. Specifically, we refer to the conventional prompt setups as *multi-prompt generation*, which is commonly used in existing consistent T2I generation methods. The multi-prompt generation uses $N$ prompts separately for each generated frame, each sharing the same *identity prompt* and the corresponding frame prompt as $[\mathcal{P}_0; \mathcal{P}_i], i \in [1, N]$. In contrast, our *single-prompt generation* concatenates all the prompts as $[\mathcal{P}_0; \mathcal{P}_1; \dots; \mathcal{P}_N]$ for each frame generation, which we refer as the *Prompt Consolidation (PCon)*.

### 3.1.1 CONTEXT CONSISTENCY IN TEXT EMBEDDINGS

Empirically, we find that the frame prompt $\{\mathcal{P}_i \mid i = 1, \dots, N\}$ in the *single-prompt generation* setup have relatively small semantic distances among each other in the textual embedding space, whereas those across *multi-prompt generation* have comparatively larger distances. For instance, we set the identity frame $\mathcal{P}_0 =$ "A watercolor of a cute kitten" as an example. We then create $N = 5$ *frame prompts* $\{\mathcal{P}_i, i \in [1, N]\}$ as "in a garden", "dressed in a superhero cape", "wearing a collar with a bell", "sitting in a basket", and "dressed in a cute sweater", respectively. Under the multi-prompt setup, each frame is generated by the text embedding defined as $\mathcal{C}_i = \tau_\xi([\mathcal{P}_0; \mathcal{P}_i]) = [\boldsymbol{c}^{SOT}, \boldsymbol{c}^{\mathcal{P}_0}, \boldsymbol{c}^{\mathcal{P}_i}, \boldsymbol{c}^{EOT}], (i = 1, \dots, N)$, while the text embedding of the *Prompt Consolidation* in the single-prompt case is $\mathcal{C} = \tau_\xi([\mathcal{P}_0; \mathcal{P}_1; \dots; \mathcal{P}_N]) = [\boldsymbol{c}^{SOT}, \boldsymbol{c}^{\mathcal{P}_0}, \boldsymbol{c}^{\mathcal{P}_1}, \dots, \boldsymbol{c}^{\mathcal{P}_N}, \boldsymbol{c}^{EOT}]$.

To analyze the distances among the *frame prompts*, we extract $\boldsymbol{c}^{\mathcal{P}_i}$ from $\mathcal{C}_i$ for multi-prompt setup and apply t-SNE for 2D visualization (Fig. 2-left). Similarly, we extract all $\boldsymbol{c}^{\mathcal{P}_i}$ from $\mathcal{C}$ for the single-prompt setup (Fig. 2-left). As can be observed, the text embeddings of *frame prompts* under the multi-prompt setup are widely distributed in the text representation space (red dots) with an average Euclidean $L_2$ distance of 71.25. In contrast, the embeddings in the single-prompt case exhibit more compact distributions (blue dots), with a much smaller average $L_2$ distance of 46.42. We also performed a similar distance analysis on all prompt sets in our benchmark *ConsiStory+*. As shown in Fig.2-right, we can conclude a similar observation that the *frame prompts* share more similar semantic information and identity consistency within the single-prompt setup.

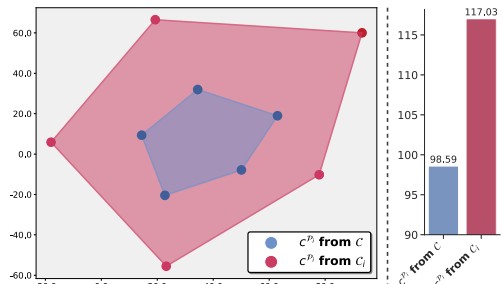

Figure 2: **t-SNE visualization of text embeddings (Left):** $\boldsymbol{c}^{\mathcal{P}_i}$ from *single-prompt generation* are closer together compared to those from *multi-prompt generation*. **Statistical results (Right):** We evaluated the average distances between the corresponding point sets of all prompt sets on the *ConsiStory+* benchmark after dimensionality reduction. The average distance between text embeddings from *single-prompt generation* is smaller than that from *multi-prompt generation*.

### 3.1.2 CONTEXT CONSISTENCY IN IMAGE GENERATION

To demonstrate that *context consistency* is also maintained in the image space, we further conducted image generation experiments using the prompt example above. The images generated by the SDXL model with the multi-prompt configuration, as illustrated in Fig. 3 (left, the first row), show various characters that lack identity consistency. Instead, we use our proposed concatenated prompt $\mathcal{P} = [\mathcal{P}_0; \mathcal{P}_1; \dots; \mathcal{P}_N]$. To generate the $i$-th frame ($i = 1, ..., N$), we reweight the $\boldsymbol{c}^{\mathcal{P}_i}$ corresponding to the desired frame prompt $\mathcal{P}_i$ by a magnification factor while rescaling the embeddings of the other *frame prompts* by a reduction factor. This modified text embedding is then imported to the T2I model to generate the frame image. We refer to this simplistic reweighting approach as *Naive Prompt Reweighting* (NPR). By this means, the T2I model synthesizes frame images with the same subject identity. However, the backgrounds get blended among these frames, as shown in Fig. 3 (left, the second row). By contrast, our full model *1Prompt1Story* introduced in Sec. 3.2 generates images with better consistent identity and text-image alignment for each frame prompt, as shown in Fig. 3 (left, the last row).

To visualize identity similarity among images, we removed backgrounds using CarveKit (Selin, 2023) and extracted visual features with DINO-v2 (Oquab et al., 2023; Darcet et al., 2023). These features are then projected into the 2D space by t-SNE (Hinton & Roweis, 2002) (as shown in Fig. 3 (mid)). Our complete approach *1Prompt1Story* obviously obtains better identity consistency than

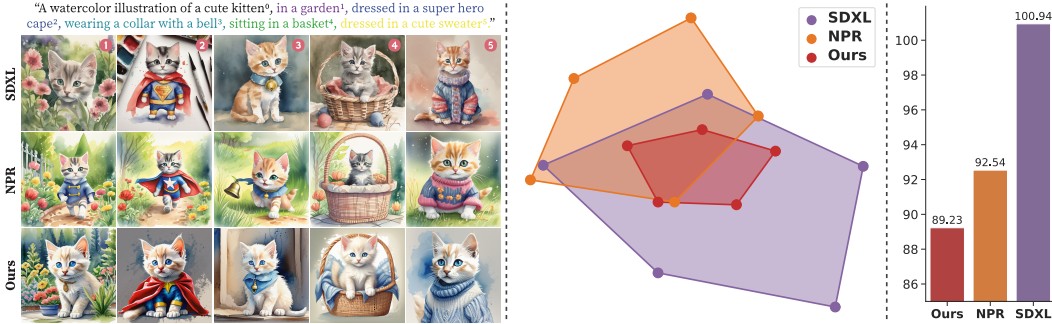

Figure 3: **(Left):** SDXL generates frame images using multi-prompt generation, while *Naive Prompt Reweighting* (NPR) and our method utilize the single-prompt setup. **(Mid):** Image features are extracted by DINO-v2 (Oquab et al., 2023) and visualized by the t-SNE reduction. *Naive Prompt Reweighting* and *1Prompt1Story* show more consistent identity generations than the SDXL model. **(Right):** Statistics of the average feature distances among generated images from the prompts in our extended *ConsiStory+* benchmark, which further confirms that *1Prompt1Story* produces better identity consistency.

the other two comparison methods, while *Naive Prompt Reweighting* shows improvements over the SDXL baseline. We also applied the analysis across our extended benchmark *ConsiStory+* and calculated the average pairwise distance, as shown in Fig. 3 (right). These results further consolidate our conclusion that the *frame prompts* in a single-prompt setup share more identity consistency than the multi-prompt case.

## 3.2 ONE-PROMPT-ONE-STORY

As also observed from the above section, simply concatenating the prompts as *Naive Prompt Reweighting* cannot guarantee that the generated images accurately reflect the frame prompt descriptions, for which we assume that the T2I model cannot accurately capture the correct partition of the concatenated prompt embeddings. Furthermore, the various semantics within the consolidated descriptions interact with each other (Chefer et al., 2023; Rassin et al., 2024). To mitigate this issue, we propose additional techniques based on the *Prompt Consolidation (PCon)*, namely *Singular-Value Reweighting* (SVR) and *Identity-Preserving Cross-Attention* (IPCA).

**Singular-Value Reweighting.** After the *Prompt Consolidation* as $\mathcal{C} = \tau_\xi([\mathcal{P}_0; \mathcal{P}_1; \ldots; \mathcal{P}_N]) = [\boldsymbol{c}^{SOT}, \boldsymbol{c}^{\mathcal{P}_0}, \boldsymbol{c}^{\mathcal{P}_1}, \ldots, \boldsymbol{c}^{\mathcal{P}_N}, \boldsymbol{c}^{EOT}]$, we require the current frame prompt to be better *expressed* in the T2I generation, which we denote as $\mathcal{P}^{exp} = \mathcal{P}_j, (j = 1, ..., N)$. We also expect the remaining frames to be *suppressed* in the generation, which we denote as $\mathcal{P}^{sup} = \mathcal{P}_k, k \in [1, N]\backslash\{j\}$. Thus, the $N$ *frame prompts* of the subject description can be written as $\{\mathcal{P}^{exp}, \mathcal{P}^{sup}\}$. As the [EOT] token contains significant semantic information (Li et al., 2023a; Wu et al., 2024b), the semantic information corresponding to $\mathcal{P}^{exp}$, in both $\mathcal{P}_j$ and [EOT], needs to be enhanced, while the semantic information corresponding to $\mathcal{P}^{sup}$, in $\mathcal{P}_k, k \neq j$ and [EOT], need to be suppressed. We extract the token embeddings for both express and suppress sets as $\mathcal{X}^{exp} = [\boldsymbol{c}^{\mathcal{P}_j}, \boldsymbol{c}^{EOT}]$ and $\mathcal{X}^{sup} = [\boldsymbol{c}^{\mathcal{P}_1}, \ldots, \boldsymbol{c}^{\mathcal{P}_{j-1}}, \boldsymbol{c}^{\mathcal{P}_{j+1}}, \ldots, \boldsymbol{c}^{\mathcal{P}_N}, \boldsymbol{c}^{EOT}]$.

Inspired by (Gu et al., 2014; Li et al., 2023a), we assume that the main singular values of $\mathcal{X}^{exp}$ correspond to the fundamental information of $\mathcal{P}^{exp}$. We then perform SVD decomposition as: $\mathcal{X}^{exp} = \boldsymbol{U}\boldsymbol{\Sigma}\boldsymbol{V}^T$, where $\boldsymbol{\Sigma} = diag(\sigma_0, \sigma_1, \cdots, \sigma_{n_j})$, the singular values $\boldsymbol{\sigma}_0 \geq \cdots \geq \boldsymbol{\sigma}_{n_j}$ [2]. To enhance the expression of the frame $\mathcal{P}_j$, we introduce the augmentation for each singular value, termed as **SVR+** and formulated as:

$$\hat{\sigma} = \beta e^{\alpha\sigma} * \sigma. \tag{2}$$

where the symbol $e$ is the exponential, $\alpha$ and $\beta$ are parameters with positive numbers. We recover the tokens as $\hat{\mathcal{X}}^{exp} = \boldsymbol{U}\hat{\boldsymbol{\Sigma}}\boldsymbol{V}^T$, with the updated $\hat{\boldsymbol{\Sigma}} = diag(\hat{\sigma_0}, \hat{\sigma_1}, \cdots, \hat{\sigma_{n_j}})$. The new prompt embedding is defined as $\hat{\mathcal{X}}^{exp} = [\hat{\boldsymbol{c}}^{\mathcal{P}_j}, \hat{\boldsymbol{c}}^{EOT}]$, and $\hat{\mathcal{C}} = [\boldsymbol{c}^{SOT}, \boldsymbol{c}^{\mathcal{P}_0}, \cdots, \hat{\boldsymbol{c}}^{\mathcal{P}_j}, \cdots, \boldsymbol{c}^{\mathcal{P}_N}, \hat{\boldsymbol{c}}^{EOT}]$. Note that there is an updated $\hat{\mathcal{X}}^{sup} = [\boldsymbol{c}^{\mathcal{P}_1}, \ldots, \boldsymbol{c}^{\mathcal{P}_{j-1}}, \boldsymbol{c}^{\mathcal{P}_{j+1}}, \ldots, \boldsymbol{c}^{\mathcal{P}_N}, \hat{\boldsymbol{c}}^{EOT}]$.

---

[2] $n_j = \min(D, |\boldsymbol{c}^{\mathcal{P}_j}| + |\boldsymbol{c}^{EOT}|)$. The dimension $D$ in the SDXL model is greater than $|\boldsymbol{c}^{\mathcal{P}_j}| + |\boldsymbol{c}^{EOT}|$)

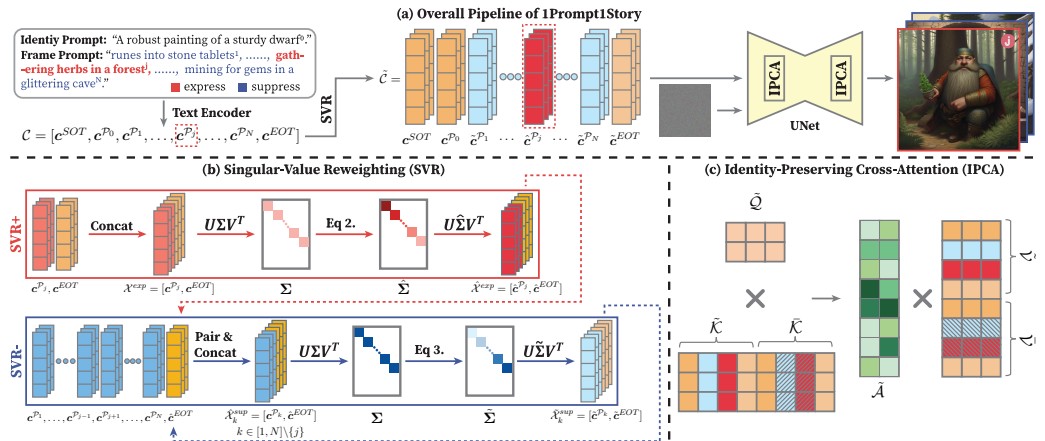

Figure 4: **(a):** The overall pipeline of *1Prompt1Story*. We combine the *identity prompt* and *frame prompts* into a single prompt, then we apply both *Singular-Value Reweighting* (SVR) and *Identity-Preserving Cross-Attention* (IPCA) to generate identity-consistent images. **(b):** During *SVR*, we first enhance the semantic information of the *express set* $\mathcal{X}^{exp}$ (red arrow), then iteratively weaken the semantics for the *suppress set* $\mathcal{X}^{sup}$ (blue arrow). **(c):** In *IPCA*, we concatenate $\tilde{\mathcal{K}}$ with $\bar{\mathcal{K}}$ and $\tilde{\mathcal{V}}$ with $\bar{\mathcal{V}}$ to improve identity consistency.

Similarly, we suppress the expression of the remaining frames. Since $\hat{\mathcal{X}}^{sup}$ contains information related to multiple frames, the main singular values of SVD in $\hat{\mathcal{X}}^{sup}$ only capture a small portion of these descriptions, which may lead to insufficient weakening of such semantics (as shown in the Appendix of Fig. 11-right). Therefore, we propose to weaken each frame prompt in $\hat{\mathcal{X}}^{sup}$ separately. We construct the matrix as $\hat{\mathcal{X}}_k^{sup} = [\boldsymbol{c}^{\mathcal{P}_k}, \hat{\boldsymbol{c}}^{EOT}], k \neq j$ to perform SVD with the singular values $\hat{\boldsymbol{\sigma}}_0 \geq \cdots \geq \hat{\boldsymbol{\sigma}}_{n_k}$. Then, each singular value is weakened as follows, termed as **SVR-**:

$$\tilde{\sigma} = \beta' e^{-\alpha'\hat{\sigma}} * \hat{\sigma}. \tag{3}$$

where $\alpha'$ and $\beta'$ are parameters with positive numbers. The recovered structure is $\tilde{\mathcal{X}}_k^{sup} = [\tilde{\boldsymbol{c}}^{\mathcal{P}_k}, \tilde{\boldsymbol{c}}^{EOT}]$, After reducing the expression of each suppress token, we finally obtain the new text embedding $\tilde{\mathcal{C}} = [\boldsymbol{c}^{SOT}, \boldsymbol{c}^{\mathcal{P}_0}, \tilde{\boldsymbol{c}}^{\mathcal{P}_1}, \cdots, \hat{\boldsymbol{c}}^{\mathcal{P}_j}, \cdots, \tilde{\boldsymbol{c}}^{\mathcal{P}_N}, \tilde{\boldsymbol{c}}^{EOT}]$.

***Identity-Preserving Cross-Attention.*** The use of *Singular-Value Reweighting* can reduce the blending of frame descriptions in *single-prompt generation*. However, we observed that it could also impact *context consistency* within the single prompt, leading to images generated slightly less similar in identity (as shown in the ablation study of Fig. 7). Recent work (Liu et al., 2024) demonstrated that cross-attention maps capture the characteristic information of the token, while self-attention preserves the layout information and the shape details of the image. Inspired by this, we propose *Identity-Preserving Cross-Attention* to further enhance the identity similarity between images generated from the concatenated prompt of our proposed *Prompt Consolidation*.

For a specific timestep $t$, after applying *Singular-Value Reweighting*, we have the updated text embedding $\tilde{\mathcal{C}}$. During a denoising pass through the diffusion model, we obtain the corresponding $\tilde{\mathcal{Q}}, \tilde{\mathcal{K}}, \tilde{\mathcal{V}}$ in the cross-attention layer. Here, we aim to strengthen the identity consistency among the images and mitigate the impact of irrelevant prompts. We set the token features in $\tilde{\mathcal{K}}$ corresponding to $\mathcal{P}_i, i \in [1, N]$ to zero, resulting in $\bar{\mathcal{K}}$. Here, only the *identity prompt* remains to augment the identity semantics. Similarly, we can get $\bar{\mathcal{V}}$. We form a new version of $\tilde{\mathcal{K}}$ by concatenating it with $\bar{\mathcal{K}}$, dubbed $\tilde{\mathcal{K}} = \texttt{Concat}(\tilde{\mathcal{K}}^{\top}, \bar{\mathcal{K}}^{\top})^{\top}$. The new cross-attention map is then given by:

$$\tilde{\mathcal{A}} = softmax\left(\tilde{\mathcal{Q}}\tilde{\mathcal{K}}^{\top}/\sqrt{d}\right) \tag{4}$$

where $d$ is the dimension of $\tilde{\mathcal{Q}}$ and $\tilde{\mathcal{K}}$. Similarly, we update $\tilde{\mathcal{V}} = \texttt{Concat}(\tilde{\mathcal{V}}^{\top}, \bar{\mathcal{V}}^{\top})^{\top}$. The final output feature of the cross-attention layer is $\tilde{\mathcal{A}} \times \tilde{\mathcal{V}}$. This output is a reweighted version that strengthens identity consistency using filtered features, which only contain the *identity prompt* semantics.

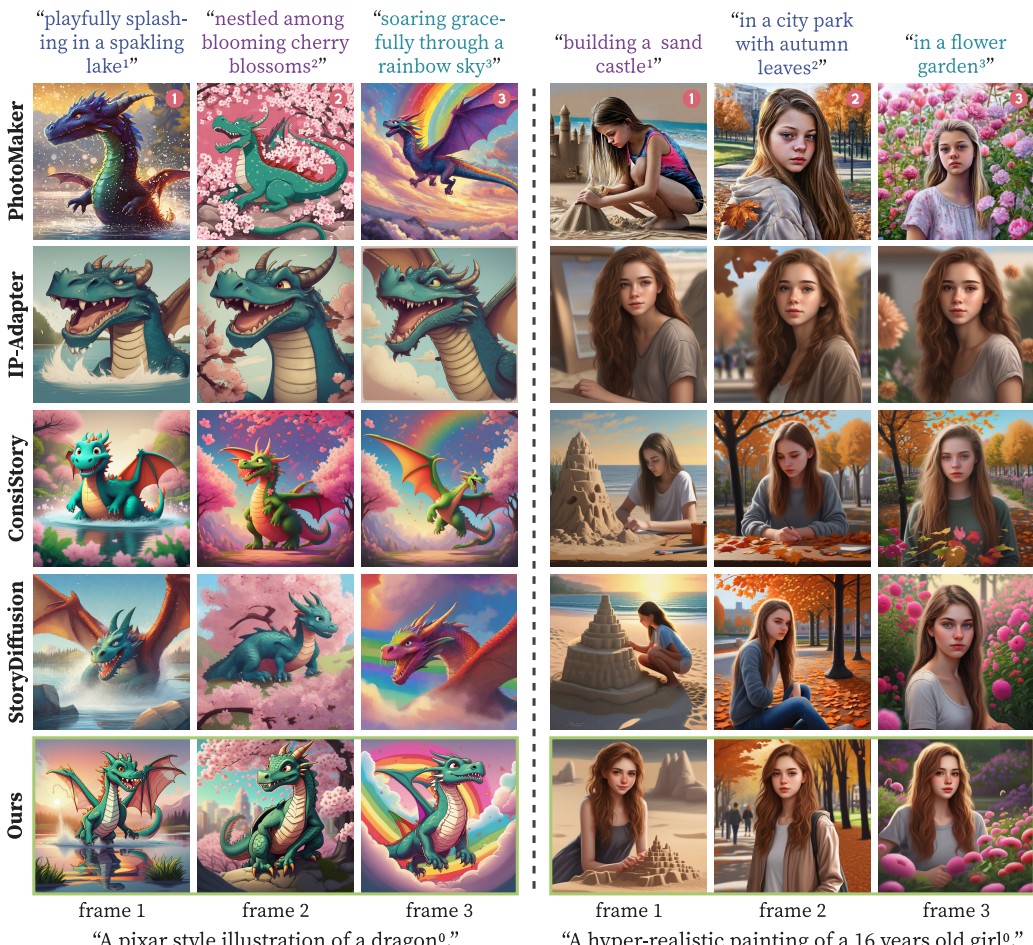

Figure 5: **Qualitative results**. We compare our method with PhotoMaker, IP-Adapter, ConsiStory, and Story-Diffsion. Among them, Texture Inversion, PhotoMaker, ConsiStory, and StoryDiffsion struggled to maintain identity consistency for the *dragon* object while IP-Adapter produced images with relatively similar poses and backgrounds. See Comparison with the remaining methods in Fig. 22 of the Appendix.

## 4 EXPERIMENTS

### 4.1 EXPERIMENTAL SETUPS

**Comparison Methods and Benchmark.** We compare our method with the following consistent T2I generation approaches: BLIP-Diffusion (Li et al., 2024), Textual Inversion (TI)(Gal et al., 2023a), IP-Adapter(Ye et al., 2023), PhotoMaker (Li et al., 2023b), The Chosen One (Avrahami et al., 2023), ConsiStory (Tewel et al., 2024), and StoryDiffusion (Zhou et al., 2024). We follow the default configurations in their papers or open-source implementations.

To evaluate their performance, we introduce *ConsiStory+*, an extension of the original ConsiStory (Tewel et al., 2024) benchmark. This new benchmark incorporates a wider range of subjects, descriptions, and styles. Following the evaluation protocol outlined in ConsiStory, we evaluated both *prompt alignment* and *subject consistency* across *ConsiStory+*, generating up to 1500 images on 200 prompt sets. Additional details on the construction of our benchmark and the implementation of the methods are provided in Appendix B.2 and Appendix B.3.

**Evaluation Metrics.** To assess *prompt alignment* performance, we compute the average CLIP-Score (Hessel et al., 2021) for each generated image in relation to its corresponding prompt, which we denote as CLIP-T. For the *identity consistency* evaluation, we measure image similarity using

Table 1: **Quantitative comparison.** The best and second best results are highlighted in **bold** and underlined, respectively. Vanilla SD1.5 and Vanilla SDXL are shown as references and excluded from this comparison.

| Method | Base Model | Train-Free | CLIP-T↑ | CLIP-I↑ | DreamSim↓ | Steps | Memory (GB)↓ | Inference Time (s)↓ |
|---|---|---|---|---|---|---|---|---|
| Vanilla SD1.5 | - | - | 0.8353 | 0.7474 | 0.5873 | 50 | 4.73 | 2.4657 |
| Vanilla SDXL | - | - | 0.9074 | 0.8165 | 0.5292 | 50 | 16.04 | 13.0890 |
| BLIP-Diffusion | SD1.5 | ✗ | 0.7607 | 0.8863 | 0.2830 | 26 | 7.75 | 1.9284 |
| Textual Inversion | | ✗ | 0.8378 | 0.8229 | 0.4268 | 40 | 32.94 | 282.507 |
| The Chosen One | SDXL | ✗ | 0.7614 | 0.7831 | 0.4929 | 35 | 10.93 | 11.2073 |
| PhotoMaker | | ✗ | 0.8651 | 0.8465 | 0.3996 | 50 | 23.79 | 18.0259 |
| IP-Adapter | | ✗ | 0.8458 | **0.9429** | **0.1462** | 30 | 19.39 | 13.4594 |
| ConsiStory | | ✓ | 0.8769 | 0.8737 | 0.3188 | 50 | 34.55 | 34.5894 |
| StoryDiffusion | SDXL | ✓ | 0.8877 | 0.8755 | 0.3212 | 50 | 45.61 | 25.6928 |
| *Naive Prompt Reweighting* (NPR) | | ✓ | 0.8411 | 0.8916 | 0.2548 | 50 | 16.04 | 17.2413 |
| *1Prompt1Story* (Ours) | | ✓ | **0.8942** | 0.9117 | 0.1993 | 50 | 18.70 | 23.2088 |

both DreamSim (Fu et al., 2023), which has been shown to closely reflect human judgment in evaluating visual similarity, and CLIP-I (Hessel et al., 2021), calculated by the cosine distance between image embeddings. In line with the methodology proposed in DreamSim (Fu et al., 2023), we remove image backgrounds using CarveKit (Selin, 2023) and replace them with random noise to ensure that similarity measurements focus solely on the identities of subjects.

## 4.2 EXPERIMENTAL RESULTS

**Qualitative Comparison.** In Fig. 5, we present the qualitative comparison results. Our method *1Prompt1Story* demonstrates well-balanced performance in several key aspects, including identity preservation, accurate frame descriptions, and diversity in the pose of objects. In contrast, other methods exhibit shortcomings in one or more of these aspects. Specifically, PhotoMaker, ConsiStory, and StoryDiffusion all produce inconsistent identities for the subject "dragon" in the examples on the left. Additionally, IP-Adapter tends to generate images with repetitive poses and similar backgrounds, often neglecting frame prompt descriptions. ConsiStory also displays duplicated background generation in the consistent T2I generation.

**Quantitative Comparison.** In Table 1, we illustrate the quantitative comparison with other approaches. In all evaluation metrics, *1Prompt1Story* ranks first among the training-free methods, and second when including training-required methods. Furthermore, compared to other training-free methods, our approach demonstrates a reasonable fast inference speed while achieving excellent performance. More specifically, our method *1Prompt1Story* achieves the CLIP-T score closely aligned with the vanilla SDXL model. In terms of identity similarity, measured by CLIP-I and DreamSim, our method ranks just below IP-Adapter. However, the high identity similarity of IP-Adapter mainly stems from its tendency to generate images with characters depicted in similar poses and layouts. To further explore this potential bias, we conducted a user study to investigate human preferences. Following ConsiStory, we also visualized our quantitative results using a chart, as shown in Fig. 6. Training-based methods, such as IP-Adapter and Textual Inversion, often overfit character identity and perform poorly on prompt alignment. In contrast, among training-free methods, our approach achieves the best balance in both prompt alignment and identity consistency.

**User Study.** In the user study, we compare our method with several state-of-the-art approaches, including IP-Adapter, ConsiStory, and StoryDiffusion. From our benchmark, we randomly selected 30 sets of prompts, each comprising four fixed-length prompts, to generate test images. Twenty participants were asked to select the image that best demonstrated overall performance in terms of identity consistency, prompt alignment, and image diversity. As shown in Table 2, the results indicated that our method *1Prompt1Story* aligns best with human preference. More details of the user study are shown in Appendix. F.

**Ablation study.** We performed an ablation study to analyze each component, as illustrated both qualitatively and quantitatively in Fig. 7 and Table 3. When using *Singular-Value Reweighting* exclusively with improving the express set as **SVR+** (that is, Eq. 2), the generated images blend with other descriptions, as can be seen in Fig. 7 (left, first row). Similarly, when *Singular-Value Reweighting* is only to weaken the suppress set as **SVR-** (i.e., Eq.3), the same issue appears in Fig. 7

Table 2: *User study* with 37 people to vote for the best consistent T2I generation method according to human preference.

| Method | IP-Adapter | ConsiStory | StoryDiffusion | Ours |
|---|---|---|---|---|
| Percent (%)↑ | 8.60 | 13.00 | 29.80 | **48.60** |

Table 3: **Ablation study.** We evaluated the influence of each component in *1Prompt1Story*, including the *Singular-Value Reweighting* (SVR+ and SVR-), and *Identity-Preserving Cross-Attention* (IPCA).

| Method | CLIP-T↑ | CLIP-I↑ | DreamSim↓ |
|---|---|---|---|
| PCon; SVR+ | 0.8774 | 0.8886 | 0.2560 |
| PCon; SVR- | 0.8910 | 0.8904 | 0.2605 |
| PCon; SVR+; SVR- | **0.8989** | 0.8849 | 0.2538 |
| PCon; SVR+; SVR-; IPCA (Ours) | 0.8942 | **0.9117** | **0.1993** |

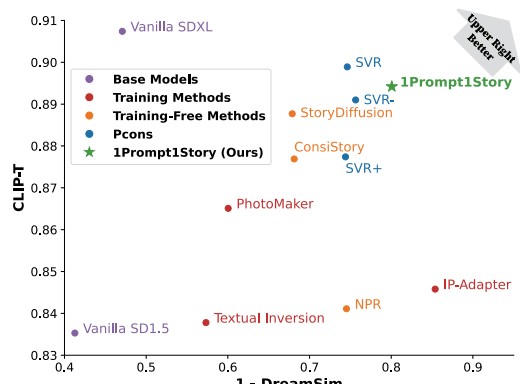

Figure 6: *Prompt alignment vs. identity consistency.* Our method *1Prompt1Story* is positioned in the upper right corner.

"A photo of a dog[0], chasing a frisbee in a colorful park[1], dancing to music at a vibrant street festival[2], jumping through a hoop at a circus performance[3], posing for a photoshoot in a modern art gallery[4]."

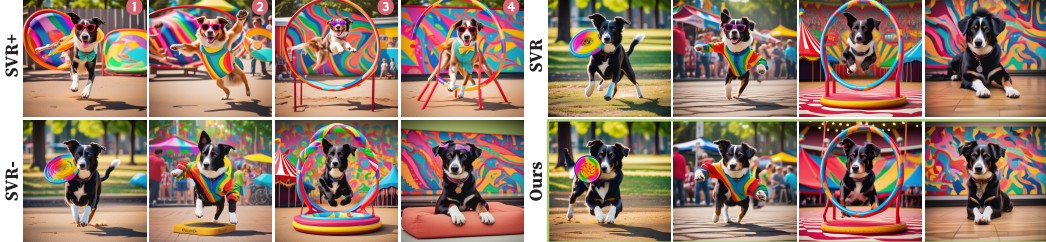

Figure 7: **Qualitative ablation study.** All ablated cases with incomplete components of *1Prompt1Story* struggle to achieve both prompt alignment and identity consistency as effectively as our full method.

(left, second row). In contrast, integrating both **SVR+** and **SVR-** in *Singular-Value Reweighting* can effectively mitigate blending in generated images (Fig. 7 (right, first row)). Although *Singular-Value Reweighting* can effectively resolve frame prompt blending issues, without *Identity-Preserving Cross-Attention*, there remains a weak inconsistency among the generated images. As shown in Fig. 7 (right, second row), the results indicate that using *Singular-Value Reweighting* and *Identity-Preserving Cross-Attention* achieves the best performance, as also evident in Table 3 (the last row). Additional results of ablation analysis and visualization are presented in the Appendix. C.

**Additional applications.** *1Prompt1Story* can also achieve spatial controls, integrating with existing control-based generative methods such as ControlNet (Zhang & Agrawala, 2023). As shown in Fig. 8 (left), our method effectively generates consistent characters with human pose control. Furthermore, our method can be combined with other approaches, such as PhotoMaker (Li et al., 2023b), to improve the consistency of identity with real images. By applying our method, the generated images more closely resemble the real identities, as demonstrated in Fig. 8 (right).

# 5 CONCLUSION

In this paper, we addressed the critical challenge of maintaining subject consistency in text-to-image (T2I) generation by leveraging the inherent property of *context consistency* found in natural language. Our proposed method, *One-Prompt-One-Story* (*1Prompt1Story*), effectively utilizes a single extended prompt to ensure consistent identity representation across diverse scenes. By integrating techniques such as *Singular-Value Reweighting* and *Identity-Preserving Cross-Attention*, our approach not only refines frame descriptions but also strengthens the consistency at the attention level. The experimental results on the *ConsiStory+* benchmark demonstrated the superiority of *1Prompt1Story* over state-of-the-art techniques, showcasing its potential for applications in animation, interactive storytelling, and video generation. Ultimately, our contributions highlight the importance of understanding context in T2I diffusion models, paving the way for more coherent and narrative-consistent visual output.

ACKNOWLEDGEMENTS

This work was supported by NSFC (NO. 62225604) and Youth Foundation (62202243). We acknowledge the support of the project PID2022-143257NB-I00, funded by the Spanish Government through MCIN/AEI/10.13039/501100011033 and FEDER. Additionally, we recognize the "Science and Technology Yongjiang 2035" key technology breakthrough plan project (2024Z120). The computations for this paper were facilitated by the resources provided by the Supercomputing Center of Nankai University (NKSC).

We would like to extend our gratitude to all the co-authors for their invaluable assistance and insightful suggestions throughout this work. In particular, we wish to thank Kai Wang, a postdoctoral researcher at the Computer Vision Center, Universitat Autònoma de Barcelona. His meticulous advice and guidance were instrumental in the completion of this project.

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

# APPENDIX

## A    BOARDER IMPACTS AND LIMITATIONS

**Boarder Impacts.** The application of T2I models in consistent image generation offers extensive potential for various downstream applications, enabling the adaptation of images to different contexts. In particular, synthesizing consistent characters has diverse applications, however, it is a challenging task for diffusion models. Our *1Prompt1Story* can help the users customize their desired characters in different story scenarios, resulting in significant time and resource savings. Notably, current methods have inherent limitations, as discussed in this paper. However, our model can serve as an intermediary solution while offering valuable insights for further advancements.

**Limitations.** While our method *1Prompt1Story* can achieve high-fidelity consistent T2I generation, it is not free of limitations. Firstly, we have to know all the prompts in advance. Additionally, the length of the input prompt is constrained by the maximum capacity of the text encoder. Although we proposed a sliding window technique that facilitates infinite-length story generation in Appendix D.2, this approach may encounter issues where the identity of the generated images gradually diverges and becomes less consistent.

## B    IMPLEMENTATION DETAILS

### B.1    MODEL CONFIGURATIONS

We generate subject-consistent images by modifying text embeddings and cross-attention modules at inference time, without any training or optimization processes. Our primary base model is the pre-trained Stable Diffusion XL (SDXL)[3]. SDXL has two text encoders: the CLIP L/14 encoder (Radford et al., 2021) and the OpenCLIP bigG/14 encoder (Cherti et al., 2023). We separately update the text embeddings produced by each encoder. For *Naive Prompt Reweighting*, we multiply the text embedding corresponding to the frame prompt that needs to be expressed by a factor of 2, while the text embedding corresponding to the *frame prompts* that need to be suppressed is multiplied by a factor of 0.5, keeping the $c^{EOT}$ unchanged.

In our method, *1Prompt1Story*, we set the parameters as follows: $\alpha = 0.01, \beta = 0.05$ in Eq.2, and $\alpha' = 0.01, \beta' = 1.0$ in Eq.3. During the generation process, we initialize all frames with the same noise and apply a dropout rate of 0.5 to the token features in $\bar{\mathcal{K}}$ corresponding to $\mathcal{P}_0$. In the implementation of IPCA, the concatenated $\tilde{\mathcal{K}}$ and $\tilde{\mathcal{V}}$ are derived from the original text embeddings prior to applying SVR. We design an attention mask where all values in the column corresponding to $\mathcal{P}_i, i \in [1, N]$ are set to zero, while all other positions are set to one. The natural logarithm of this mask is then added to the original attention map. Our full algorithm is presented in Algorithm 1. Following (Tewel et al., 2024; Alaluf et al., 2024; Luo et al., 2023), we use Free-U (Si et al., 2024) to enhance the generation quality. All generated images based on SDXL are produced at a resolution of $1024 \times 1024$ using a Quadro RTX 3090 GPU with 24GB VRAM.

### B.2    BENCHMARK DETAILS

To evaluate the effectiveness of our method, we developed *ConsiStory+*, an extended prompt benchmark based on ConsiStory (Tewel et al., 2024). We enhanced both the diversity and size of the original benchmark, which only comprised 100 sets of 5 prompts across 4 superclasses. Our expansion resulted in 200 sets, with each set containing between 5 and 10 prompts, categorized into 8 superclasses: humans, animals, fantasy, inanimate, fairy tales, nature, technology, and foods. The extended prompt benchmark was generated using ChatGPT 4.0-turbo[4], involving two main steps. First, we expanded the 100 prompt sets from the original benchmark, increasing each to a length of 5 to 10 prompts, as shown in Fig. 9 (left). Then, we generated new prompt sets for each of the new superclasses, as illustrated in Fig. 9 (right). The prompt sets collected through these two steps were combined to form our benchmark, *ConsiStory+*.

---

[3]https://huggingface.co/stabilityai/stable-diffusion-xl-base-1.0
[4]https://chatgpt.com/

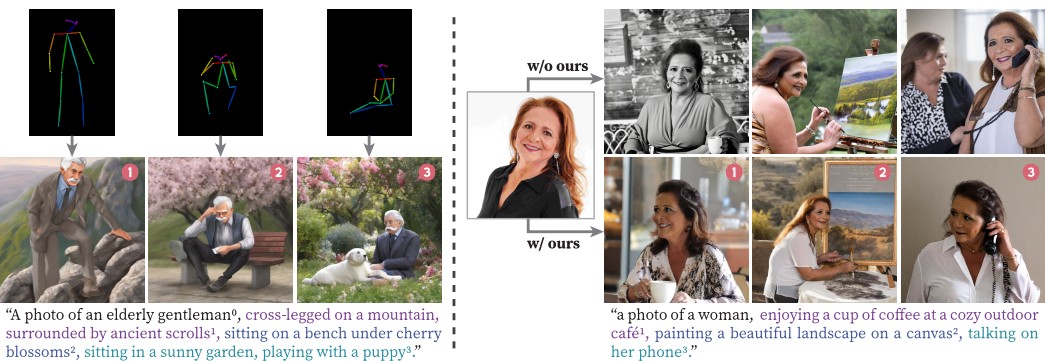

"A photo of an elderly gentleman[0], cross-legged on a mountain, surrounded by ancient scrolls[1], sitting on a bench under cherry blossoms[2], sitting in a sunny garden, playing with a puppy[3]."

"a photo of a woman, enjoying a cup of coffee at a cozy outdoor café[1], painting a beautiful landscape on a canvas[2], talking on her phone[3]."

Figure 8: **(Left):** Our method *1Prompt1Story* can integrate with ControlNet to enable spatial control for consistent character generation. **(Right):** Additionally, our method can also combine with other methods, such as PhotoMaker, to achieve real-image personalization with improved identity consistency.

---

**Algorithm 1** *1Prompt1Story*

**Input** : A text embedding $\mathcal{C} = [\boldsymbol{c}^{SOT}, \boldsymbol{c}^{\mathcal{P}_0}, \boldsymbol{c}^{\mathcal{P}_1}, \cdots, \boldsymbol{c}^{\mathcal{P}_N}, \boldsymbol{c}^{EOT}]$ and latent vector $z_t$.
**Output:** The subject consistency images $\mathcal{I}_1, \cdots, \mathcal{I}_N$.

---

**for** $j = 1, \ldots, N$ **do**
    // Singular-Value Reweighting
    $\hat{\mathcal{X}}^{exp} = [\hat{\boldsymbol{c}}^{\mathcal{P}_j}, \hat{\boldsymbol{c}}^{EOT}] \leftarrow \mathcal{X}^{exp} = [\boldsymbol{c}^{\mathcal{P}_j}, \boldsymbol{c}^{EOT}]$ (Eq. 2);
    **for** $k = [1, N] \setminus \{j\}$ **do**
        $\tilde{\mathcal{X}}^{sup} = [\tilde{\boldsymbol{c}}_k^{\mathcal{P}}, \tilde{\boldsymbol{c}}^{EOT}] \leftarrow [\boldsymbol{c}_k^{\mathcal{P}}, \hat{\boldsymbol{c}}^{EOT}]$ (Eq. 3);
    **end**
    $\tilde{\mathcal{C}} = [\boldsymbol{c}^{SOT}, \boldsymbol{c}^{\mathcal{P}_0}, \tilde{\boldsymbol{c}}^{\mathcal{P}_1}, \cdots, \hat{\boldsymbol{c}}^{\mathcal{P}_j}, \cdots, \tilde{\boldsymbol{c}}^{\mathcal{P}_N}, \tilde{\boldsymbol{c}}^{EOT}]$;

    // Identity-Preserving Cross-Attention
    **for** $t = T, \ldots, 1$ **do**
        $\tilde{\mathcal{K}}, \tilde{\mathcal{V}} \leftarrow \tilde{\mathcal{C}}$;
        $\bar{\mathcal{K}}, \bar{\mathcal{V}} \leftarrow \tilde{\mathcal{K}}, \tilde{\mathcal{V}}$;
        $\tilde{\mathcal{K}} = \texttt{Concat}(\tilde{\mathcal{K}}^\top, \bar{\mathcal{K}}^\top)^\top, \ \tilde{\mathcal{V}} = \texttt{Concat}(\tilde{\mathcal{V}}^\top, \bar{\mathcal{V}}^\top)^\top$;
        $\tilde{\mathcal{A}} \leftarrow \tilde{\mathcal{Q}}, \tilde{\mathcal{K}}$ (Eq. 4);
        $z_{t-1} \leftarrow \epsilon_\theta(z_t, t, \tilde{\mathcal{C}})$ with $\tilde{\mathcal{A}}, \tilde{\mathcal{V}}$;
    **end**
    $\mathcal{I}_j = D(z_0)$
**end**
**Return** $\mathcal{I}_1, \cdots, \mathcal{I}_N$.

---

## B.3 COMPARISON METHOD IMPLEMENTATIONS

We compare our method with all other approaches based on Stable Diffusion XL, except for BLIP-Diffusion (Li et al., 2024), which is based on Stable Diffusion v1.5[5]. The DDIM steps is set to the default value in the open-source code of each method. Below are the third-party packages we used for method implementations:

- The unofficial implementation of Textual Inversion (TI) (Gal et al., 2023a) at `https://github.com/oss-roettger/XL-Textual-Inversion`.
- The unofficial implementation of The Chosen One (Avrahami et al., 2023) at `https://github.com/ZichengDuan/TheChosenOne`.
- The official implementation of IP-Adapter (Ye et al., 2023) at `https://github.com/tencent-ailab/IP-Adapter`.
- The official implementation of PhotoMaker (Li et al., 2023b) at `https://github.com/TencentARC/PhotoMaker`.

---

[5] https://huggingface.co/runwayml/stable-diffusion-v1-5

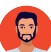

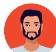

{
      "prompt": "A photo of A sticker of a cute corgi dog, on a student's laptop, amidst an action-packed skate session, showing its teeth, at a live music gig, at an international airport.",
      "concept_token": "dog",
      "superclass": "inanimate",
      "subject": "A sticker of a cute corgi dog",
      "settings": [
          "on a student's laptop",
          "amidst an action-packed skate session",
          "adding character to urban signage",
          "at a live music gig",
          "at an international airport"
      ],
      "style": "A photo of"
}
According to the template above, **select a random number between 5 and 10, and extend the settings to that length.** The extended content should describe the subject. Also, update the content in the prompt accordingly.

{
      "prompt": "A photo of A sticker of a cute corgi dog, on a student's laptop, amidst an action-packed skate session, showing its teeth, at a live music gig, at an international airport.",
      "concept_token": "dog",
      "superclass": "inanimate",
      "subject": "A sticker of a cute corgi dog",
      "settings": [
          "on a student's laptop",
          "amidst an action-packed skate session",
          "adding character to urban signage",
          "at a live music gig",
          "at an international airport"
      ],
      "style": "A photo of"
}
According to the template above, **replace the superclass category with 'fair_tales' and choose a new random style, subject, and settings based on this superclass.** Ensure that the concept_token represents the subject described, and the settings' length is a random number between 5 and 10, describing various actions of the subject.

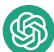

Here is the updated version based on the instructions:
{
      "prompt": "A photo of A sticker of a cute corgi dog, on a student's laptop, amidst an action-packed skate session, adding character to urban signage, at a live music gig, at an international airport, brightening up a park bench, stuck to a water bottle in a hiking trail.",
      "concept_token": "dog",
      "superclass": "inanimate",
      "subject": "A sticker of a cute corgi dog",
      "settings": [
          "on a student's laptop",
          "amidst an action-packed skate session",
          "adding character to urban signage",
          "at a live music gig",
          "at an international airport",
          "brightening up a park bench",
          "stuck to a water bottle in a hiking trail"
      ],
    "style": "A photo of"
}
**I extended the settings to a random length of 7** and updated the prompt accordingly.

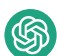

Here is the updated version based on the instructions:
{
      "prompt": "An illustration of A mischievous elf, dancing in the moonlight, weaving through the forest trees, whispering to the woodland creatures, playing a flute by the riverbank, hiding behind a mushroom.",
      "concept_token": "elf",
      "superclass": "fair_tales",
      "subject": "A mischievous elf",
      "settings": [
          "dancing in the moonlight",
          "weaving through the forest trees",
          "whispering to the woodland creatures",
          "playing a flute by the riverbank",
          "hiding behind a mushroom"
      ],
      "style": "An illustration of"
}
In this update, the superclass was changed to "fair_tales," and **a new subject, style, and settings were chosen to fit this theme. The settings' length was randomly selected to be 5.**

Figure 9: **(Left):** We expand the length of the original prompt sets to a random number between 5 and 10. **(Right):** We generate a new prompt set within one of the new superclass "fairy tales".

- The official implementation of BLIP-Diffusion (Li et al., 2024) at `https://github.com/salesforce/LAVIS/tree/main/projects/blip-diffusion`.
- The official implementation of StoryDiffusion (Zhou et al., 2024) at `https://github.com/HVision-NKU/StoryDiffusion`.

Since Consistory (Tewel et al., 2024) is not open-source, we reimplemented it ourselves. During the inference time, BLIP-Diffusion (Li et al., 2024), IP-Adapter (Ye et al., 2023), and PhotoMaker (Li et al., 2023b) all require a reference image as the additional input. To generate the reference image, we use their corresponding base models, providing the identity description as the input prompt. For example, if the full prompt is "a photo of a beautiful girl walking on the street", we use "a photo of a beautiful girl" to generate the reference image. The reference image is then used to generate all frames in the corresponding prompt set.

**Identity Prompt:** "A photo of a fox[0]."

**Frame Prompt:** ❶ "wearing a scarf in a meadow[1]" ❷ "playing in the snow[2]" ❸ "at the edge of a river[3]"

Order: ❶ ❷ ❸                     Order: ❶ ❸ ❷

Order: ❷ ❶ ❸                     Order: ❷ ❸ ❶

Order: ❸ ❶ ❷                     Order: ❸ ❷ ❶

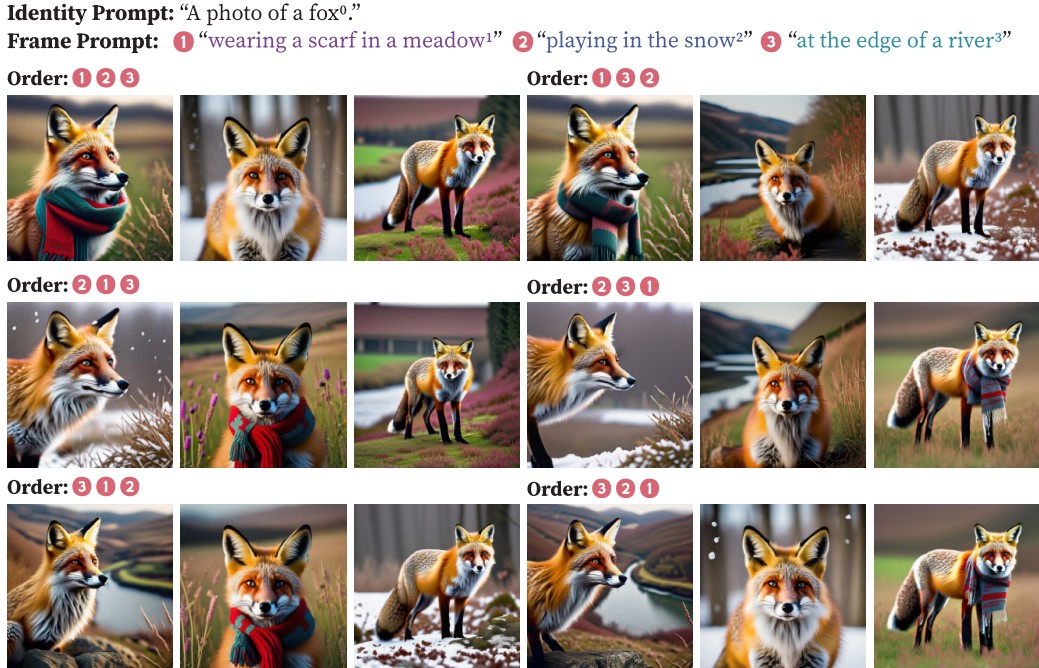

Figure 10: **Robustness to frame prompts order.** With the same set of *frame prompts* but in different orders, our method *1Prompt1Story* consistently generates images with a unified identity.

## C    ADDITIONAL ABLATION STUDY

### C.1    ROBUSTNESS TO DIVERSE DESCRIPTION ORDERS

To validate the robustness of our method regarding the order of *frame prompts*, we used the same three *frame prompts*: "wearing a scarf in a meadow", "playing in the snow", and "at the edge of a river" to create six different sequences for images generation. The *identity prompt* was consistently set to "a photo of a fox" and each sequence used the same seed for a generation. As shown in Fig. 10, our method *1Prompt1Story* generates images with identity consistency across different orders. Furthermore, the content of the images generated from varying sequences is closely aligned with the text descriptions, further demonstrating our method *Singular-Value Reweighting* effectiveness in suppressing content of unrelated *frame prompts*.

### C.2    *Singular-Value Reweighting* ANALYSIS

Our *Singular-Value Reweighting* algorithm comprises two successive components: SVR+ enhances the *frame prompts* we wish to express, while SVR- iteratively weakens the *frame prompts* we aim to suppress. In our experiments, we first apply SVR+, followed by SVR-. In particular, we found that performing SVR- before SVR+ also yields similar results (see Fig. 11-left).

In the process of applying SVR-, we employed a strategy of iteratively suppressing each frame prompt. In fact, we could also concatenate the text embeddings corresponding to all frame prompts for suppression. To explore this, we conducted further ablation study specifically on the SVR-component. Assuming that we have $n$ frames to generate, we discovered that merging the text embeddings corresponding to the $n-1$ frames we wish to suppress with $c^{\mathrm{EOT}}$ and subsequently performing the SVD decomposition does not effectively extract the main components of all *frame prompts* included in $c^{\mathrm{EOT}}$. Consequently, applying Eq. 3 to weaken the eigenvalues based on their magnitude fails to adequately eliminate the descriptions of all suppressed frames. we refer to this as "joint suppress", as illustrated in Fig. 11 (right, the first row). In contrast, if we handle each frame prompt to be suppressed individually and iteratively perform SVD and the operations from Eq. 3, which we term "iterative suppress", we can more effectively suppress all irrelevant *frame prompts*, as shown in Fig. 11 (right, the second row).

"A mystical illustration of a wise wizard[0], casting a spell under a full moon[1], standing before a portal in the forest[2], in a tower filled with ancient artifacts[3], summoning a storm over a mountain peak[4]."

"A mischievous goblin[0], climbing the walls of a deserted castle[1], laughing by a flickering campfire[2], running through an underground tunnel[3], scurrying through a dense forest at twilight[4]."

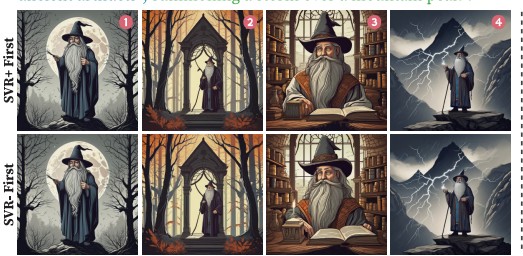
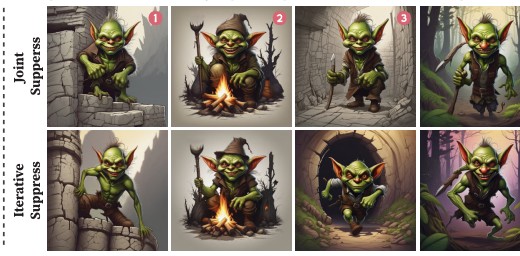

Figure 11: **(Left):** "SVR+ First" indicates that SVR+ is applied before SVR- in the *Singular-Value Reweighting* process, while "SVR- First" means the opposite order. We found that both sequences yield similar results (same seed). **(Right):** Compared to "Joint Suppress", "Iterative Suppress" is more effective at minimizing the influence of other *frame prompts* when generating images for the current frame. "Joint Suppress" produces images with similar backgrounds (the first row, first and third columns).

"A watercolor illustration of A puppy[0], wearing a small sweater[1], digging a hole[2], wearing a training harness[3], sticking head out of the window[4], swimming[5], playing in the yard[6]."

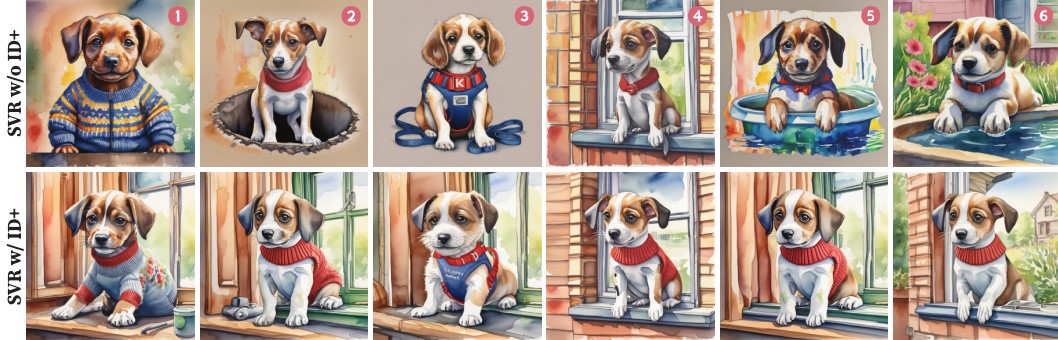

Figure 12: **SVR with identity enhancement.** The first row represents the original SVR with enhancements applied only to the frame prompt. The second row builds upon the original by further enhancing the identity prompt in the SVR+ module. The results indicate that while the second method improves identity consistency, it also leads to more similar object poses and backgrounds.

In our SVR, we enhance only the current frame prompt that needs to be expressed. An alternative option is to enhance the identity prompt simultaneously. We found that doing so can make the object's identity more consistent; however, it also introduces some negative effects, the background and subject's pose appearing more similar across images, as shown in Fig. 12. Furthermore, to demonstrate the role of the $c^{\text{EOT}}$ in SVR, we conducted an ablation study on the $c^{\text{EOT}}$ component. Specifically, we kept the $c^{\text{EOT}}$ part of the text embedding unchanged during the SVR process and used this text embedding to generate images. As shown in Fig. 13, the results indicate that without performing SVR on the $c^{\text{EOT}}$, the backgrounds of different frame prompts tend to blend together.

### C.3 *Naive Prompt Reweighting* ABLATION STUDY

Similar to the *Singular-Value Reweighting* (SVR) experiment, we conducted an ablation study to verify the effectiveness of *Naive Prompt Reweighting* (NPR) in terms of identity preservation and prompt alignment compared to our method *1Prompt1Story*. We denote NPR+ as applying a scaling factor of 2 to the text embedding corresponding to the current frame prompt that needs to be expressed. Conversely, NPR- denotes applying a scaling factor of 0.5 to the text embeddings of all other *frame prompts* that need to be suppressed. NPR represents the combination of both NPR+ and NPR- operations.

As shown in Fig. 14, images generated using the NPR+, NPR-, and NPR methods all exhibit varying degrees of interference from other *frame prompts*. In contrast, our method effectively removes

"A digital portrait of a 14-year-old boy[0], in a rose garden[1], making a sandcastle[2], playing in snow[3]."

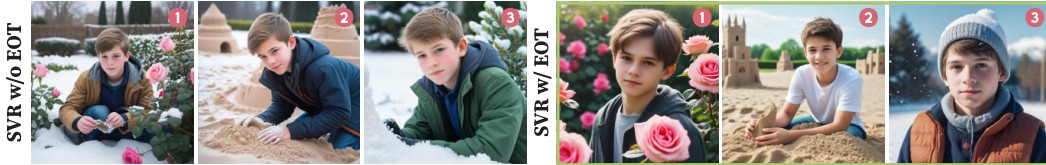

Figure 13: **Ablation study for $c^{\text{EOT}}$.** The left three images demonstrate the SVR process with a fixed $c^{\text{EOT}}$, while the right illustrates the SVR procedure described in the main text. The results indicate that keeping $c^{\text{EOT}}$ unchanged leads to background blending across images generated for different frame prompts, highlighting the importance of updating $c^{\text{EOT}}$ dynamically.

"A watercolor illustration of a male child[0], in a toy store[1], exploring an exhibit[2], in a backyard, playing with a puppy[3], enjoying a carousel ride[4], building a fort[5]."

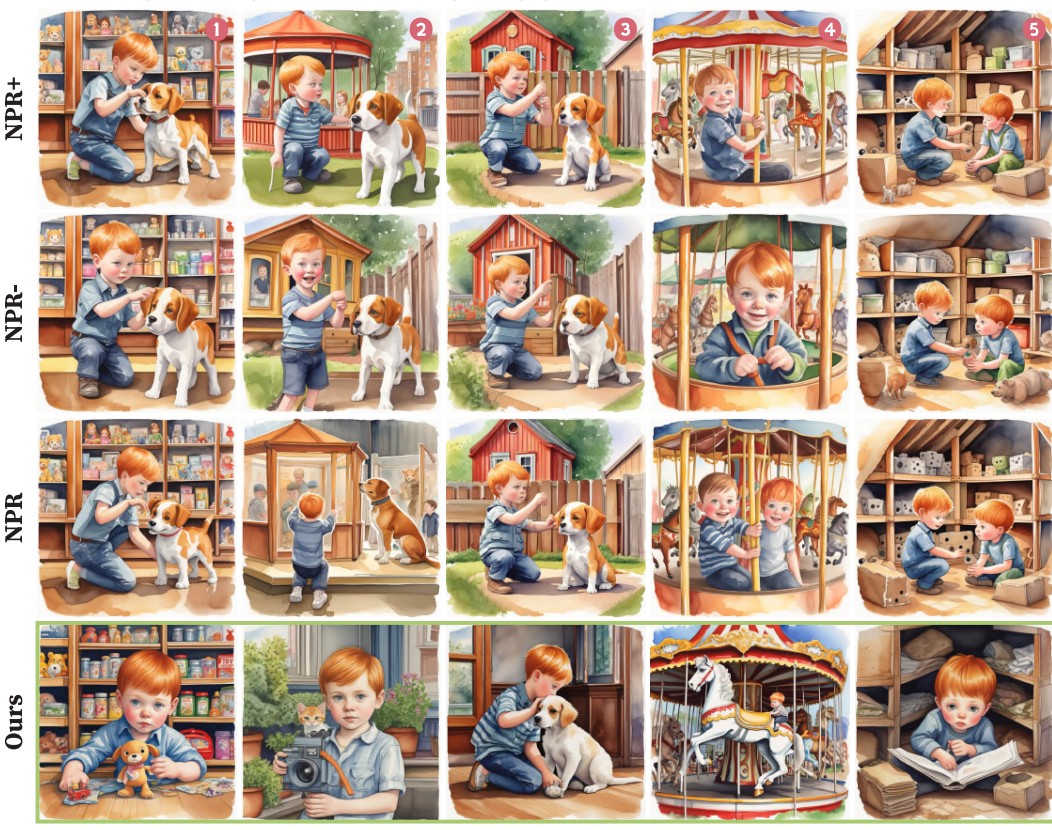

Figure 14: *Naive Prompt Reweighting* ablation study. NPR+, NPR-, and NPR are ineffective at suppressing the influence of other *frame prompts*. For example, the "puppy", which appears only in the frame prompt of the third frame, also shows up in the first and second frames using the aforementioned methods. In contrast, our method (the last row) effectively suppresses unwanted semantic information from other *frame prompts*.

irrelevant semantic information from other frame subject descriptions in the single-prompt setting, resulting in images that are more aligned with their corresponding *frame prompts*.

## C.4 SEED VARIETY

Since our method *1Prompt1Story* does not modify the original parameters of the diffusion model, it preserves the inherent ability of the model to generate images with diverse identities and backgrounds using different seeds. By varying the initial noise while keeping the input prompt set constant, our method can produce a range of characters and backgrounds, all while maintaining strong identity consistency and prompt alignment, as shown in Fig. 15.

"A hyper-realistic digital painting of an elderly gentleman[0], wearing a smoking jacket[1], at a vintage car show[2], wearing a vineyard owner's attire[3], on a golf course[4], at a classical music concert[5], painting a landscape[6]."

Figure 15: **Seed variation.** By using different seeds, our method *1Prompt1Story* can generate images with diverse backgrounds while maintaining a consistent identity.

## D  ADDITIONAL RESULTS OF OUR METHOD *1Prompt1Story*

### D.1  CONSISTENT STORY GENERATION WITH MULTIPLE SUBJECTS.

Our method is capable of generating stories involving multiple subjects. By specifying several subjects in the *identity prompt* and appending corresponding *frame prompts*, we can directly produce a series of images that maintain consistent identities across these subjects, as demonstrated in Fig. 16. However, this approach has a limitation: all generated images will include every character referenced in the *identity prompt*, which poses a constraint on the flexibility of our method.

### D.2  STORY GENERATION OF ANY LENGTH.

To generate stories of any length, we designed a "sliding window" technique to overcome the input text length limitations of diffusion models like SDXL. Suppose we aim to generate a story with $n$ images, each corresponding to $n$ *frame prompts*, using a window size $t$, where $t < n$. Similarly, we represent the *identity prompt* as $\mathcal{P}_0$ and the *frame prompts* as $\mathcal{P}_i$, where $i \in [1, n]$. For generating the image corresponding to the $i$-th frame, if $i \leq t$, we use $\mathcal{P} = [\mathcal{P}_0; \mathcal{P}_1; \ldots; \mathcal{P}_t]$ as input prompt and apply our method *1Prompt1Story* to generate the images. If $i > t$, we use $\mathcal{P} = [\mathcal{P}_0; \mathcal{P}_{i-t+1}; \ldots; \mathcal{P}_i]$ to generate the images. As shown in Fig. 19, we applied an ultra-long prompt to generate 42 images with consistent identities, using a window size of 10.

"A photo of a happy **hedgehog** with its **cheese**[0], amid blooming spring flowers[1], beside a sparkling stream[2], peeking from a cozy burrow[3], in an autumn forest[4], next to a tiny cheese wheel[5], sitting on a mushroom[6]."

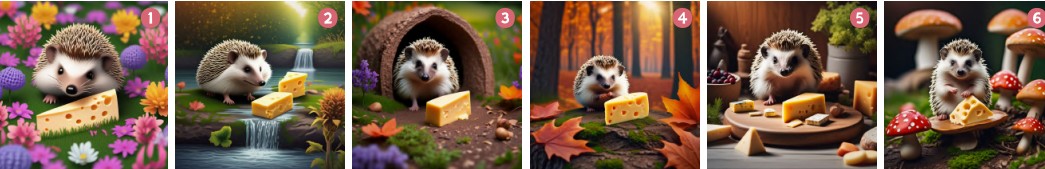

"A hyper-realistic digital painting of a young ginger **boy** with his **ball**[0], by an old brick wall covered in colorful graffiti[1], in the middle of a street filled with cars[2], near a bustling playground[3], next to a lake reflecting the early morning light[4], set against the backdrop of sunset[5], standing in a quiet meadow, under a cloudy sky[6]."

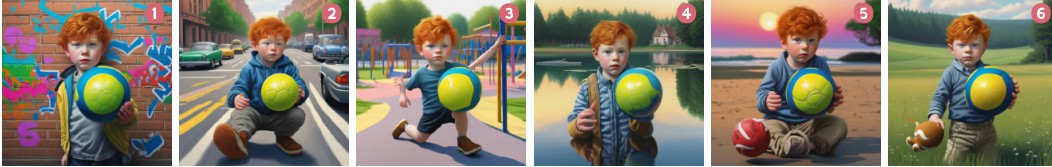

"A cinematic portrait of a **man** and a **woman**[0], in a cozy coffee shop with large windows[1], walking along a sandy beach at sunset[2], on a bustling city street at night[3], on a quiet park bench amidst falling leaves[4], under an umbrella during a soft rain[5], in a vibrant art gallery surrounded by paintings[6]."

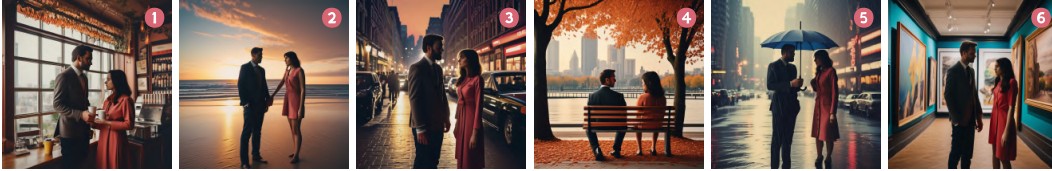

Figure 16: **Multi-subject story generation.** By defining multiple subjects in the *identity prompt*, our method generates images featuring multiple characters, each maintaining good identity consistency.

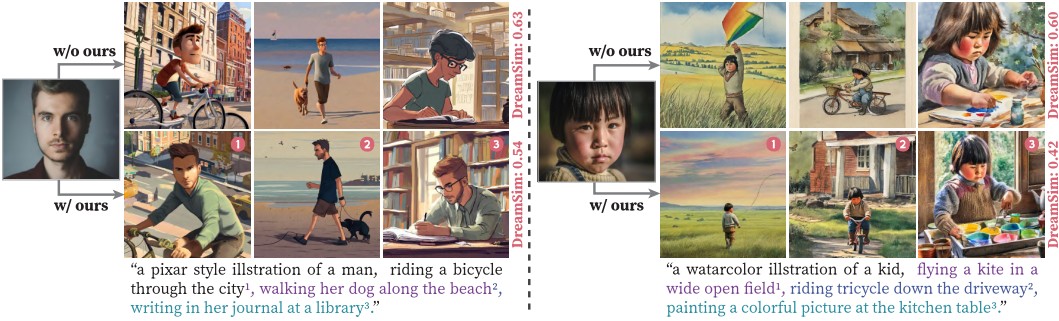

Figure 17: **Additional result with PhotoMaker.** We compared additional results of our method combined with PhotoMaker, where a lower DreamSim score indicates better ID consistency between the generated images. The results demonstrate that our method has the potential to enhance the performance of PhotoMaker.

### D.3 COMBINE WITH DIFFERENT DIFFUSION MODELS.

Since our method exclusively modifies the text-embedding and cross-attention modules of the diffusion model, it can be directly adapted to other diffusion models. In this study, we implemented our approach within the SDXL framework. Other models utilizing the SDXL framework, such as playground-v2.5[6], RealVisXL_V4.0[7] and Juggernaut-X-v10[8], can apply our method without any additional modifications or fine-tuning. Our experimental results (see Fig. 20) indicate that these models can also achieve image generation with enhanced identity consistency when employing our method *1Prompt1Story*.

---

[6] https://huggingface.co/playgroundai/playground-v2.5-1024px-aesthetic
[7] https://huggingface.co/SG161222/RealVisXL_V4.0
[8] https://huggingface.co/RunDiffusion/Juggernaut-X-v10

| Metric | SD1.5 | SDXL | BLIP-Diffusion | Textual Inversion | The Chosen One | PhotoMaker | IP-Adapter | ConsiStory | Story Diffusion | NPR | Ours |
|---|---|---|---|---|---|---|---|---|---|---|---|
| VQAScore↑ | 0.7157 | 0.8473 | 0.5735 | 0.6655 | 0.6990 | 0.8178 | 0.7834 | 0.8184 | **0.8335** | 0.8044 | 0.8275 |
| DSG w/ dependency↑ | 0.7354 | 0.8524 | 0.6128 | 0.7219 | 0.6667 | 0.8108 | 0.7564 | 0.8196 | 0.8400 | 0.8407 | **0.8520** |
| DSG w/o dependency↑ | 0.8095 | 0.8961 | 0.6909 | 0.8051 | 0.7495 | 0.8700 | 0.8122 | 0.8696 | 0.8853 | 0.8863 | **0.8945** |
| FID↓ | - | - | 65.32 | 48.94 | 83.74 | 55.27 | 66.76 | 45.20 | 51.63 | **44.02** | 44.16 |

Table 4: **Additional metircs comparison.** SD1.5 and SDXL are shown as references and excluded from this comparison. The **bold** and underlined are the best and second best results respectively.

# E    ADDITIONAL EXPERIMENTS

## E.1    ADDITIONAL PROMPT ALIGNMENT METRICS

In addition to the primary evaluation metrics, we conduct an experiment using the recent prompt alignment metrics DSG(Cho et al., 2023) and VQAScore(Lin et al., 2025). Both DSG and VQA are metrics that measure the consistency between images and text by evaluating questions and their corresponding answers. These metrics have been shown to provide more reliable strengths in fine-grained diagnosis and align closely with human judgment. We present our comparison with all other methods in Table 4, results show that our method *1Prompt1Story* outperforms other training-based methods and achieves the highest value on the DSG metric.

## E.2    VISUAL QUALITY COMPARSION

To evaluate the impact of different methods on image quality under ID consistency generation, we use images generated by the base model as the real dataset and images generated by each method itself as the fake dataset. Then, we calculate the FID(Heusel et al., 2017). As shown in Table 4 (the last row), *Naive Prompt Reweighting* (NPR) and our method *1Prompt1Story* achieved the best and second-best results in terms of FID. This indicates that our method has a smaller impact on the image generation quality of the base model compared to other methods.

## E.3    CONTEXT CONSISTENCY IN TEXT EMBEDDINGS

Besides the separate t-SNE dimensionality reduction conducted for multi-prompt and single-prompt setups in sec. 3.1.1, we extended our analysis by performing a joint t-SNE reduction on the combined text embeddings from both setups. This unified approach allows for a direct visual comparison of the embeddings' spatial arrangements within the text representation space. As illustrated in Fig. 18 (left), the text embeddings originating from the multi-prompt setup remain widely dispersed (red dots), indicative of their diverse semantic properties. Conversely, embeddings from the single-prompt setup (blue dots) exhibit noticeably tighter clustering. To substantiate these observations, we also perform statistical analysis on our benchmark dataset, as shown in Fig. 18 (right).

# F    USER STUDY DETAILS

In the user study, we compared our method with three state-of-the-art approaches: IP-Adapter, Consistory, and Story Diffusion. We selected 30 prompt sets from our *ConsiStory+* benchmark to generate test images, with each prompt set producing four frames.

In the questionnaire, participants were first provided with guidance on selecting images. They were instructed to choose the set that exhibited the most balanced performance across three criteria: identity consistency, prompt alignment, and image diversity, according to their personal preferences. As illustrated in Fig. 21, we de-

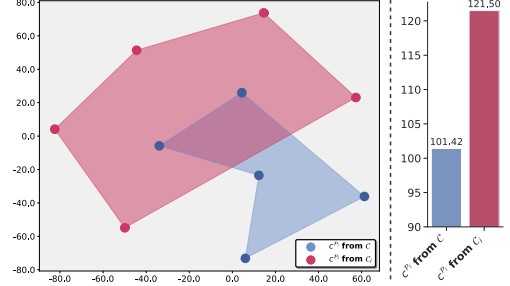

Figure 18: **Additonal t-SNE visualization of text embeddings (Left) and statistical results (Right).**

tailed these criteria at the beginning of the questionnaire. Additionally, we provided an example to demonstrate our recommended best choice, including justifications for both selecting and not selecting each set, thereby aiding participants in making informed decisions.

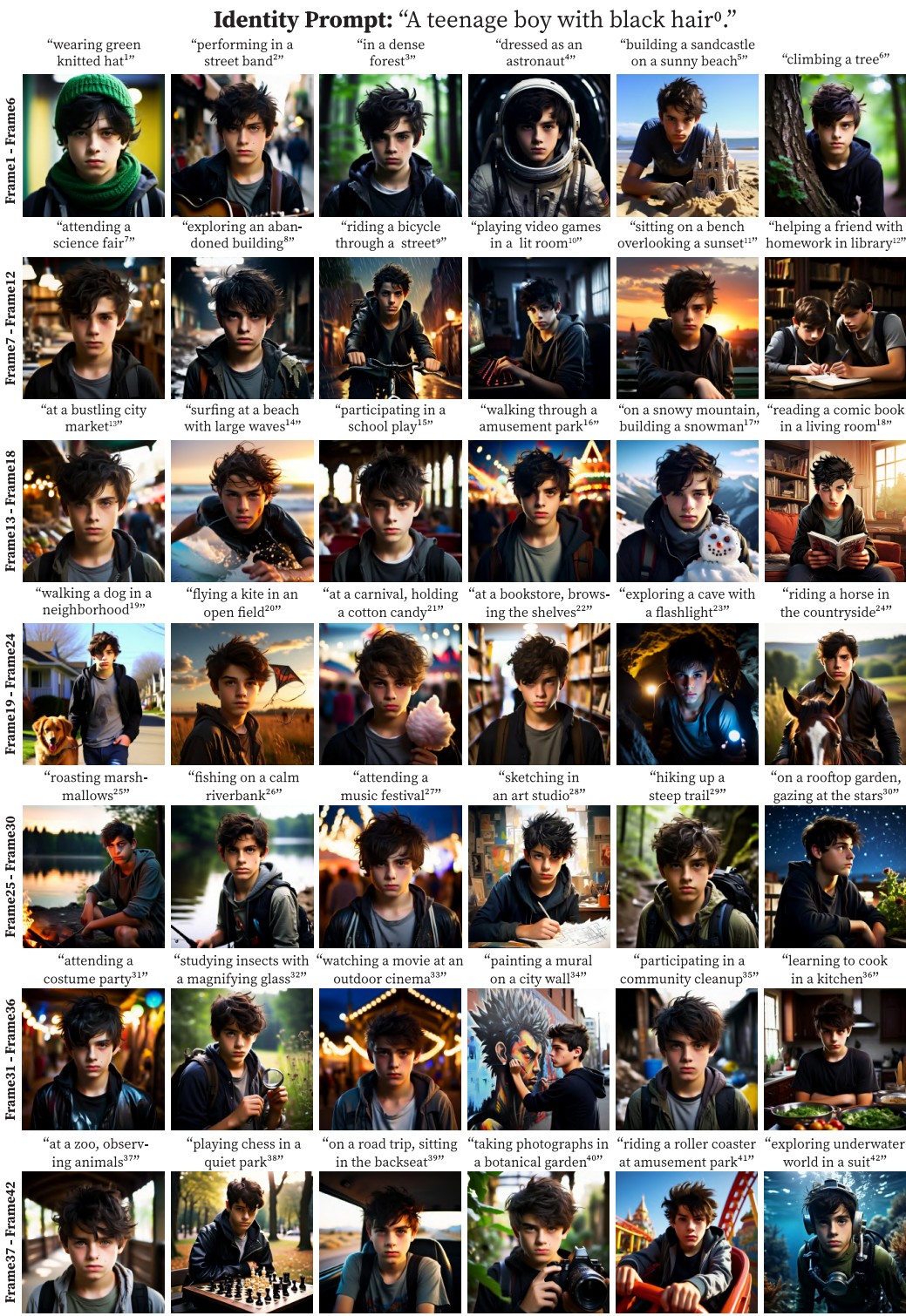

Figure 19: **Long story generation.** By using the "sliding window" technique, our method *1Prompt1Story* can generate stories of any length with consistent identity throughout.

**SDXL:** "A vintage-style poster of a **vase** with flowers[0], adding charm to homely setting[1], holding a vibrant arrangement of sunflowers[2], displaying exotic orchids[3], containing cherry blossoms[4], filled with lavender and wild daisies[5], holding bouquet of flowers[6]."

**PlayGround-v2.5:** "A photo of a **dog**[0], riding a bike on a city street[1], picking flowers in a meadow[2], eating ice cream at a park[3], drawing by a pool[4], skateboarding in a skate park[5], blowing bubbles[6]."

**RealvisXL_4.0:** "A heartwarming illustration of a friendly **troll**[0], sitting by a campfire[1], carving runes into a rock[2], building shelter from fallen logs[3], fishing in a quiet stream[4], guarding a treasure chest in dark cave[5], helping travelers across a river[6]."

**Juggernaut-X-v10:** "A quaint illustration of a **hobbit**[0], enjoying a feast under a starlit sky[1], celebrating with friends in a tavern[2], read book in a sunlit meadow[3], walking through peaceful village[4], sitting by a fireplace[5], working in a garden of vibrant vegetables[6]."

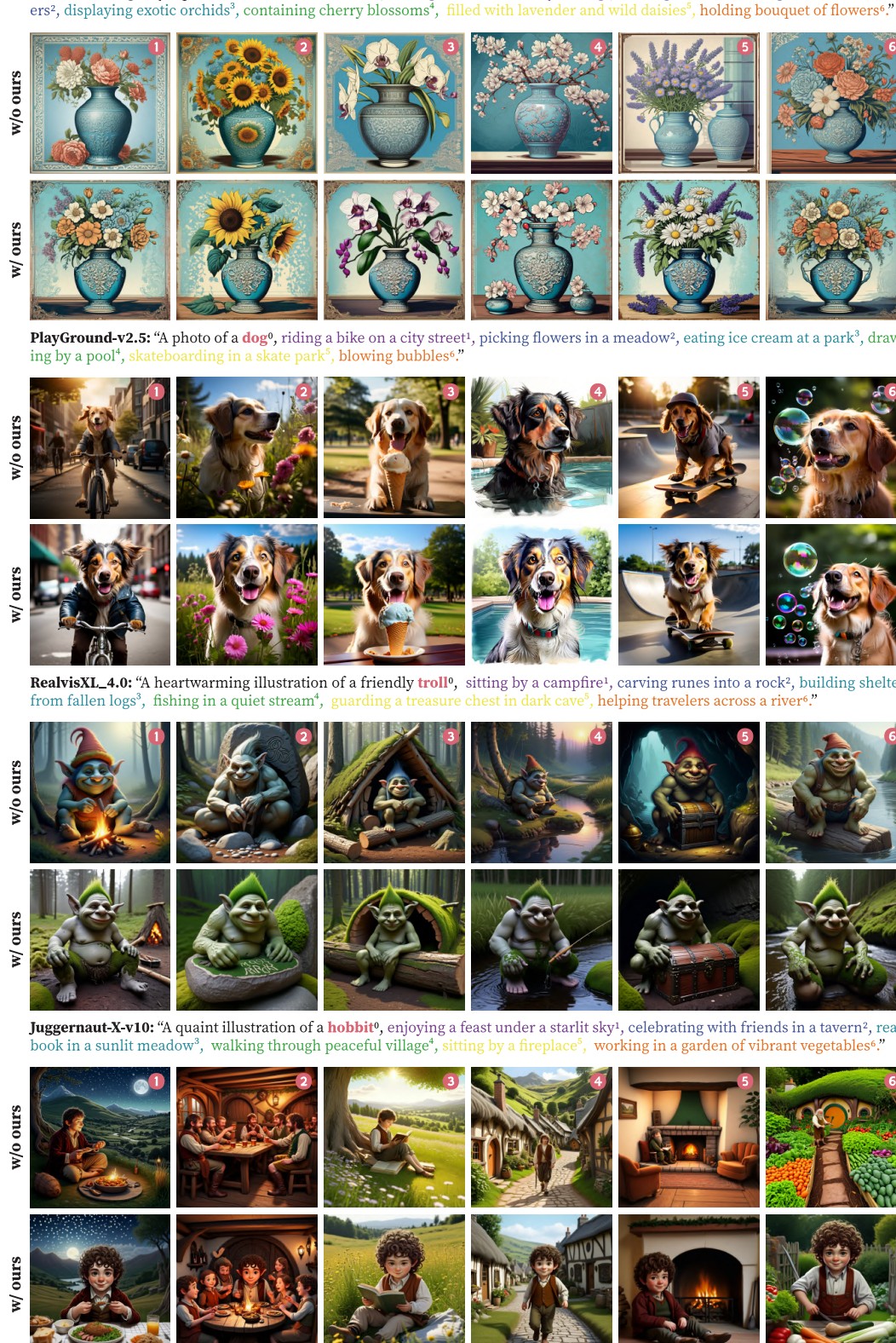

Figure 20: **Evaluation with different models.** We test our method on various T2I diffusion models, and without requiring fine-tuning, our approach could directly generate images with a consistent identity.

## Selection Guidance

In this survey, you will evaluate four sets of images based on three criteria: **"Identity Consistency" "Prompt Alignment"** and **"Image Diversity"**. Your task is to select the set that performs best across all three aspects.

**Identity Consistency:** Refers to the visual coherence of the subject's appearance across the set, indicating that the same subject is depicted in all images.

**Prompt Alignment:** Indicates how well each image in the set matches its corresponding text description.

**Image Diversity:** Refers to the variety of poses, object arrangements, and overall composition within the set of images.

## Example

Each row represents one of the four image sets: A, B, C, and D. Each column corresponds to the same frame descriptions: **['wearing a superhero cape', 'at the beach', 'wearing a headscarf', 'wearing a birthday hat'].**

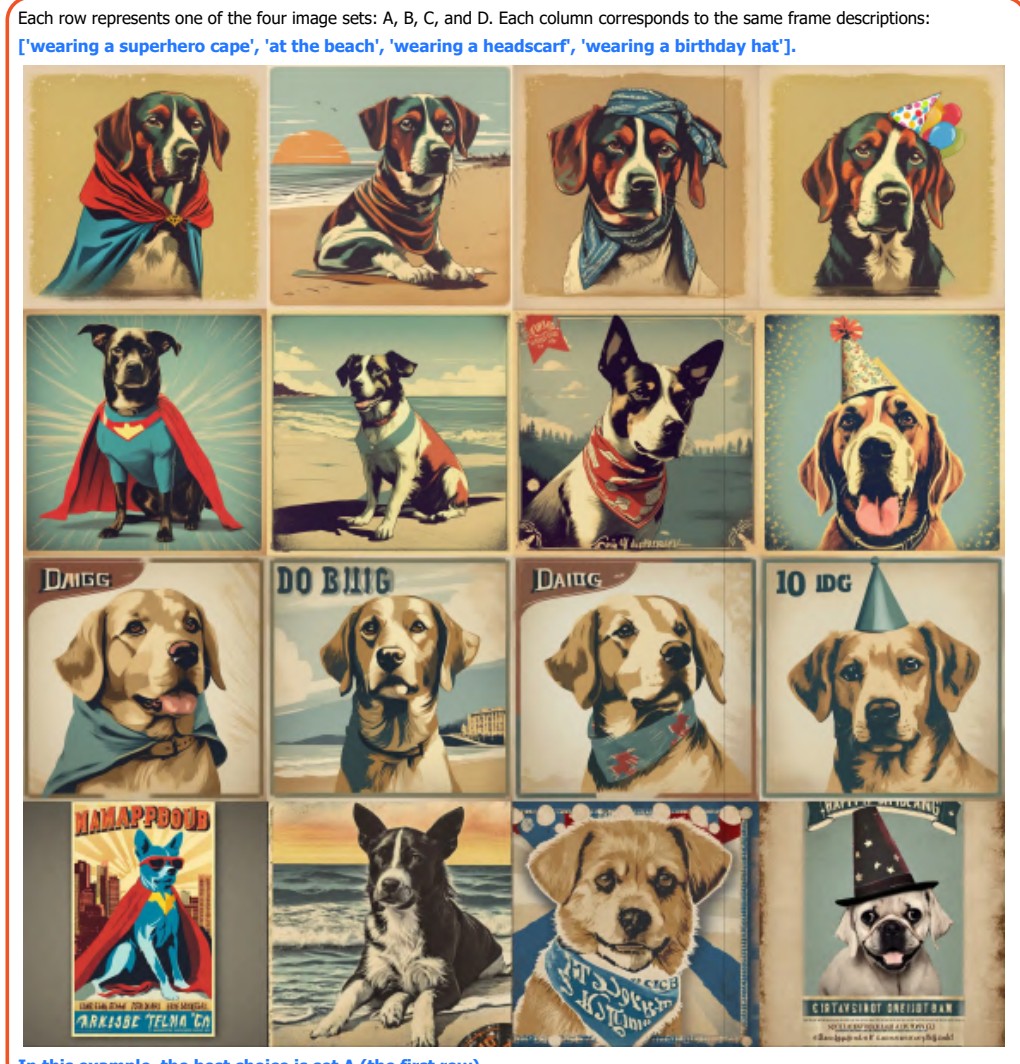

**In this example, the best choice is set A (the first row).**

## Reason for Selection

Set A (the first row) performs well in terms of "Identity Consistency," "Text Alignment," and "Image Diversity".

Set B (the second row) is not chosen because its identity consistency is poor.

Set C (the third row) is not selected despite its high identity consistency because its text alignment and image diversity are lacking.

Set D (the fourth row) is also not chosen due to its poor identity consistency.

Figure 21: **User study questionnaire.** Before filling out the questionnaire, participants were provided with selection guidelines, including detailed explanations of the three evaluation criteria: identity consistency, prompt alignment, and image diversity. Additionally, an example was provided, along with our recommended best choice and the reasoning behind the selection.

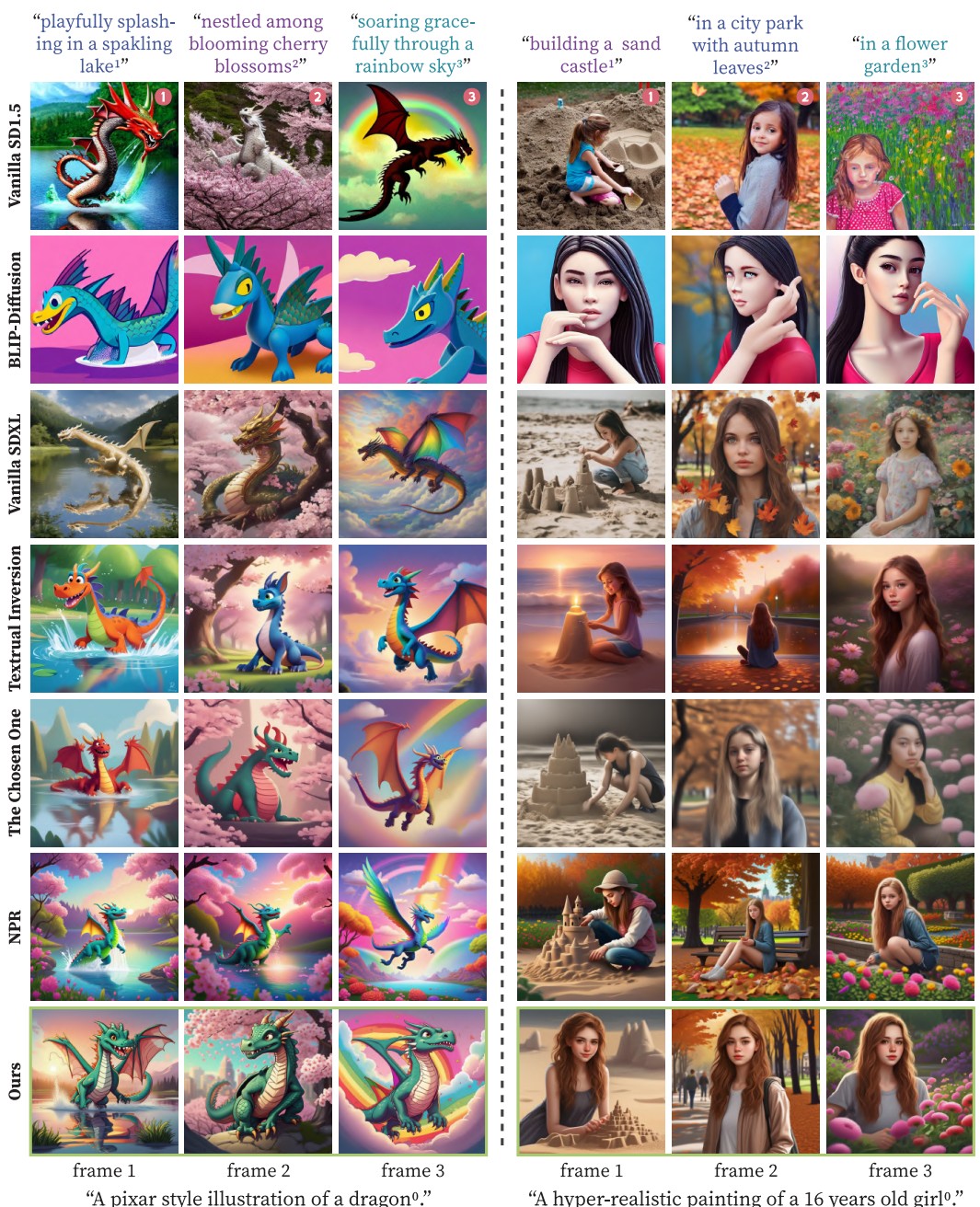

Figure 22: **Additional qualitative comparison.** We also compared our method with other existing approaches. The characters generated by vanilla SD1.5 and vanilla SDXL exhibit significant variations in both form and appearance. In contrast, some training-based methods, such as Textual Inversion and The Chosen One, generate characters with consistent forms, but their appearance lacks similarity. While NPR can produce characters with consistent identities, the backgrounds often blend across images. In contrast, our method not only ensures identity consistency but also generates backgrounds that closely align with the corresponding text descriptions.

