# OpenReview forum: "One-Prompt-One-Story: Free-Lunch Consistent Text-to-Image Generation Using a Single Prompt"
_ICLR.cc/2025/Conference — ICLR 2025 Spotlight_

### Official Review · Reviewer_7cWp · 2024-10-22

**Soundness:** 3
**Presentation:** 4
**Contribution:** 3
**Rating:** 8
**Confidence:** 5

**Summary:**

This paper introduces a novel, training-free approach called 1Prompt1Story to address the character consistency issue in storytelling. By concatenating all prompts into a single input, 1Prompt1Story preserves character identities from the start. The generation process is further refined using two innovative techniques: Singular-Value Reweighting and Identity-Preserving Cross-Attention. Comparative experiments demonstrate that 1Prompt1Story improves identity consistency and alignment with input descriptions, outperforming existing methods both quantitatively and qualitatively. Further experiments also show that 1Prompt1Story supports ControlNet.

**Strengths:**

- The authors' observation of inherent context consistency in text, along with their proposed NPR method, is particularly insightful. It offers a direct and elegant solution for maintaining consistency in text-to-image generation, making a meaningful contribution to the field.
- The paper is well-written, clearly structured, and easy to follow.
- The quantitative analysis of context consistency, particularly through t-SNE visualizations and text-prompt distance metrics, is sound and convincing.
- The authors employ a comprehensive set of metrics to demonstrate the advantages of their method, providing a robust evaluation that underscores the effectiveness of their solution.

**Weaknesses:**

- The Prompt Consolidation method concatenates all text prompts into a single input before feeding it to the text encoder. However, in my view, this approach faces a significant limitation for generating longer stories due to the token length restrictions of the diffusion model’s text encoder. In contrast, other training-free personalized storytelling methods do not encounter this issue, which limits the broader applicability of the proposed method. It would be helpful if the paper explicitly addressed the length limitation of the 1Prompt1Story approach. Additionally, are there any potential solutions to mitigate this constraint? Or will you compare the practical story length capabilities of your method versus other training-free approaches?
- The handling of the EOT tokens in lines 290-322 appears somewhat unclear. Could you further clarify why the EOT needs to be initially expressed and then suppressed? Additional explanation of this process would enhance understanding. Will you provide a brief explanation of the role of EOT tokens in the model?
- The user study sample size of 20 participants seems relatively small, which may limit the generalizability of the findings.

**Questions:**

Please the first and second point of the weaknesses section.

---

> ### Author Response · Authors · 2024-11-20
> **Response to Reviewer 7cWp**
>
> Thank you for your appreciation of our work and for your insightful feedback. Below, we outline our responses to the weaknesses and questions you brought up.
>
> **W1 & Q1: The Prompt Consolidation method concatenates all text prompts into a single input before feeding it to the text encoder. However, in my view, this approach faces a significant limitation for generating longer stories due to the token length restrictions of the diffusion model’s text encoder.**
> This is also a key challenge addressed in our method. To counter the token length limit of the CLIP model, we introduced a sliding window technique to overcome this limitation, enabling infinite-length story generation. The details of this technique are elaborated in Appendix D.2 and illustrated in Fig. 17.
>
> **W2 & Q2: Could you further clarify why the EOT needs to be initially expressed and then suppressed? Will you provide a brief explanation of the role of EOT tokens in the model?**
> The ablation study presented in Fig.11(left) of the Appendix confirms that the order of expression and suppression has minimal impact on the output, demonstrating the robustness of our approach. Additionally, the EOT token's capacity to encapsulate all preceding text aligns with observations in prior research [1,2]. By aggregating the cumulative frame prompt information, the EOT token serves as a dependable anchor for achieving consistent T2I generation. Our SVR method utilizes this property effectively, isolating and amplifying the expressed components while diminishing the suppressed ones, all while preserving coherence across frames.
>
> [1]Senmao Li, Joost van de Weijer, Fahad Khan, Qibin Hou, Yaxing Wang, et al. Get what you want, not what you don’t: Image content suppression for text-to-image diffusion models. (ICLR2024)
> [2]Yinwei Wu, Xingyi Yang, and Xinchao Wang. Relation rectification in diffusion model. (CVPR 2024)
>
> **W3: The user study sample size of 20 participants seems relatively small, which may limit the generalizability of the findings.**
> For the initial user study, we randomly selected 20 prompt sets across various categories, with each example comprising story generation images from four methods. The detailed information about the user study is introduced in Appendix.F and illustrated in Fig. 19.
>
> To enhance the reliability and breadth of the study, **we expanded it to include 37 questionnaires, each with 30 prompt sets of questions**. The updated results, summarized below, provide more robust evidence of the effectiveness of our approach across diverse scenarios.
>
> | method | 1Prompt1Story (Ours)  | StoryDiffusion | Consistory | IP-Adapter |
> | :--------: | :--------: |:--------: |:--------: |:--------: |
> | User Preference (%) | 48.6 |  29.8 |  13.0  |  8.6  |

---

> > ### Comment · Reviewer_7cWp · 2024-11-26
> >
> > Thanks for your reply. I believe this is a strong paper. I will keep my rating.

---

> > > ### Author Response · Authors · 2024-11-26
> > > **Thank you**
> > >
> > > Dear Reviewer 7cWp,
> > >
> > > Thank you very much for your kind and encouraging feedback. we deeply appreciate your thoughtful review and are genuinely grateful for your recognition of the work. Your positive remarks motivate us to continue striving to improve our research.
> > >
> > > Thank you again for your time and support.
> > >
> > > Best regards,
> > > Authors of submission 903

---

### Official Review · Reviewer_eHi5 · 2024-11-02

**Soundness:** 4
**Presentation:** 4
**Contribution:** 3
**Rating:** 8
**Confidence:** 4

**Summary:**

The paper proposes a training-free approach for consistent character generation. In particular, given a descriptive prompt of  the main character, followed by frame-specific prompts, the goal is to generate images that faithfully depict each frame, while maintaining a consistent visual appearance of the main character.

The main observation of the paper is that when combining all frame-prompts with the character description prompt, then the character embedding in each-frame prompt is more consistent. The authors introduce two additional enhancements for their final method.

[1] Singular Value Reweighting (SVR): combining all frame-prompts leads to information blending from different frames. To avoid this, during the generation of frame i (i.e., based on prompt p_i), the authors use SVD to increase p_i contribution to the generation by enhancing the leading singular value of prompt p_i. The authors also suppress the contribution of other other prompts j != i in a similar manner. Here, the authors also take into consideration the end token (EOT), as it contains significant semantic information as well.

[2] Identity-Preserving Cross-Attention: SVR alone handles intra- fame information blending, although the identity consistency may still be affected. The authors propose to concatenate the k/v- projections in which the token features corresponding to each prompt pi, i in [1,N] are zeroed out.

Minor:
L 353 (K, V) notations are different from those used later in the paragraph. Since this is a paper on consistency, please maintain consistent notations :-)

**Strengths:**

[1] The paper is well written and the quantitative and qualitative comparisons are well carried.

[2] I find the main observation of the paper to be very novel and might have broader implications to other tasks. In particular, exploiting the context consistency of language models for the purpose of character consistent generation. The paper analysis of this observation clearly motivates the rest of the paper.

[3] Using SVR for prompt-specific enhancement/suppression rather than simple reweighing is interesting.

[4] The visual quality of the generated outputs is good.

[5] The method is fairly simple and doesn’t complicated procedures

**Weaknesses:**

[1] In the Identity-Preserving Cross-Attention module, the authors zero-out features of all frame-prompts p_i. It is not clear why not completely removing these features, instead of zeroing them? As zeroing out the features may affect the attention scores, because of the normalization. Also, why the remaining features (i.e, in this case P0 features and EOT features) are enough to enhance consistent identity among frames ? Effectively, this new module (IPCA) biases queries to attend to P_0 and EOT token features. Furthermore,  P_0 and EOT embeddings are different for different frames (because of the asymmetric nature of SVR), hence the additional K/V matrices that are concatenated for different prompts are different as well, hence this means that P_0 and EOT should maintain similar embeddings for different frame-generations, which is not clear from the paper.

[2] The authors claim their method to be memory-efficient compared to existing methods. I think it is nice to know the memory complexity of the proposed method compared to existing ones.

**Questions:**

Following my points raised in the weakness section, can you please consider answering the following ?

[1] Can you please confirm that you simply zero-out the token features of all prompts P_i, i \in [1,N] ,except for P_0 and EOT ?

[2] What is the rationale behind zeroing out features of frame-prompts p_i rather than removing them entirely? How does this affect attention scores and normalization?

[3] How do the P_0 and EOT features alone enhance identity consistency across frames? Given that P0 and EOT embeddings may differ across frames due to SVR, how does your method ensure consistent embeddings for different frame generations?

[4] Can you please provide further motivation to the KV extension and why this generates more consistent characters. To me this is similar to the KV sharing used in Consistory, but here the the KV-extension only biases the queries to attended PO and EOT (which can be different for different frames due to the SVR procedure).

[6] Can you provide memory complexity comparison between your method and existing approaches.

---

> ### Author Response · Authors · 2024-11-20
> **Response to Reviewer eHi5**
>
> Thank you for recognizing our work and for your valuable feedback. Below, we offer a point-by-point response to the weaknesses and questions you raised.
>
> **Minor: L 353 (K, V) notations are different from those used later in the paragraph.**
> Thanks for your remind, we have already fixed the typo in Line 353
>
> **W1&Q1&Q2&Q3&Q4: Can you please confirm that you simply zero-out the token features of all prompts P_i, i \in [1,N] ,except for P_0 and EOT ? What is the rationale behind zeroing out features of frame-prompts p_i rather than removing them entirely? How do the P_0 and EOT features alone enhance identity consistency across frames? Can you please provide further motivation to the KV extension and why this generates more consistent characters.**
>
> Sorry for the misunderstanding and our vague description. Here we reply to the weakness point and your questions 1 to 4 in detail.
> **What we meant by “zero_out” is to set the weight of ( K ) in the attention map for ( p_i ) to zero**. In practice, we do not set the corresponding value of ( K ) to zero. Instead, we design an attention mask where all values in the column corresponding to ( p_i ) are set to zero, and other positions are set to one (for unchanging them). Then, we add its natural logarithm value to the original attention map. That effectively makes the position corresponding to ( p_i ) negative infinity, resulting in a probability of zero after softmax. In the implementation, we program it as:
>
> “attn_weights_pos = torch. softmax(scores_pos + torch.log(mask), dim=-1)”
>
> which corresponds to line 246 in utils.py in our code of the supplementary material. We already make the description clearer in the implementation details in the Appendix B.1.
>
> Regarding the P0 and EOT features, it is another point that may lead to your misunderstanding. To clarify, after SVR, the EOT feature is modified, but the P0 feature remains unchanged as we do not apply any adjustments to it. During IPCA, we concatenate P0 and EOT before applying SVR. This approach has an additional advantage: the ID information in the unmodified P0 and EOT features may more closely align with the original shared identity in a single long prompt. Such observation can be seen in the ablation study in Table 3. In practice, we employ an additional branch to retrieve the original P0 and EOT features, the corresponding implementation is detailed in the supplementary material (util.py, line 665).
>
> **W2&Q6: The authors claim their method to be memory-efficient compared to existing methods. I think it is nice to know the memory complexity of the proposed method compared to existing ones.**
> We provide a GPU memory usage comparison across all methods. Among these, "The Chosen One" is a training-based approach. In contrast, our method only requires a single concatenation operation in the cross-attention of IPCA, resulting in a negligible increase in memory consumption for the cross-attention module. On the other hand, training-free methods like Consistory and StoryDiffusion execute operations within the self-attention mechanism, which significantly increases memory usage, especially when generating more than three frames. Unlike these methods, our approach maintains a constant memory footprint regardless of the number of generated images, allowing it to operate efficiently on a standard 24GB graphics card for consistent T2I generation with numerous frames.
>
> | Method              | SD1.5 | SDXL  | BLIP-Diffusion | Textual Inversion | The Chosen One | PhotoMaker | IP-Adapter | Consistory | StoryDiffusion | NPR   | 1Prompt1Story (Ours) |
> |:-------------------:|:-----:|:-----:|:--------------:|:-----------------:|:--------------:|:----------:|:----------:|:----------:|:--------------:|:-----:|:---------------------:|
> | GPU Memory (GB)     | 4.73  | 16.04 | 7.75           | 32.94             | 10.93          | 23.79      | 19.39      | 34.55      | 45.61           | 16.04 | 18.70                 |
> | Training-Free       | -     | -     | ✗              | ✗                | ✗              | ✗          | ✗          | ✓          | ✓              | ✓     | ✓                     |

---

> > ### Comment · Reviewer_eHi5 · 2024-11-22
> >
> > Thank you for the response, I have no further questions. I will keep my original rating (8-Accept).
> > Good luck.

---

> > > ### Author Response · Authors · 2024-11-22
> > > **Thank you**
> > >
> > > Thank you very much for your thoughtful comments and kind words. We appreciate your time and effort in reviewing our work. Your feedback has been valuable to us, and we will continue to improve the quality of our research.

---

### Official Review · Reviewer_uG1g · 2024-11-03

**Soundness:** 4
**Presentation:** 4
**Contribution:** 3
**Rating:** 8
**Confidence:** 3

**Summary:**

This paper introduces a novel training-free approach for generating subject-consistent images using text-to-image models. The method concatenates prompts for different frames into a single prompt to achieve consistency, while utilizing two key innovations: singular-value reweighting and identity-preserving cross-attention techniques. These techniques effectively mitigate interference between frames and enhance consistency. Through comprehensive experiments, the authors demonstrate their method achieves superior performance in terms of text-image alignment and subject consistency, compared with both training-required and training-free competitors.

**Strengths:**

- The paper is well-written, and the results are nicely presented.
- The proposed method is simple yet effective. The paper demonstrates its effectiveness by comparing it with multiple training-required and training-free methods, along with the user study and the ablation study.
- Due to its simplicity, the proposed method should be able to easily be adapted to T2I models with different architectures and sampling strategies.

**Weaknesses:**

- Long prompts might pose a challenge to the applicability of the proposed method due to the limit of number of tokens (e.g., M=77 for SDXL by default). Nevertheless, this appears to be a solvable issue.

**Questions:**

- For the experiment in Section 3.1.1, it seems that the t-SNE visualization is conducted for only a single set of prompts. It would benefit the clarity if more details can be provided.
- Instead of the proposed Identity-Preserving Cross-Attention technique, is it possible to improve the consistency by performing SVR on $[c^{P_0}, C^{EOT}]$?
- Does the proposed method affect the visual quality/image realism? How is it compared with other methods?

---

> ### Author Response · Authors · 2024-11-20
> **Response to Reviewer uG1g**
>
> Thank you for acknowledging our work and for your thoughtful comments. Below, we provide a detailed response to the weaknesses and questions you raised.
>
> **W1: Long prompts might pose a challenge to the applicability of the proposed method due to the limit of number of tokens**
> This is also a key challenge addressed in our method. To counter the token length limit of the CLIP model, we introduced a sliding window technique to overcome this limitation, enabling infinite-length story generation. The details of this technique are elaborated in Appendix D.2 and illustrated in Fig. 17.
>
> **Q1: For the experiment in Section 3.1.1, it seems that the t-SNE visualization is conducted for only a single set of prompts. It would benefit the clarity if more details can be provided.**
> Indeed, our primary focus is on internal identity consistency within the storytelling scenario. Since diverse prompt sets are with different text embeddings and are hard to draw in a single figure. Therefore, it is logical to visualize the textual embeddings within a single prompt set under two different configurations: our single-prompt setup and the conventional multi-prompt setup. t-SNE is a non-linear dimensionality reduction method and relies on the overall context during decomposition, which makes it suitable to measure the internal prompts similarities as shown in Fig.2. **Furthermore, we present the average distance metrics in Fig. 2-right, showcasing the statistical results over the Consistory+ benchmark**.
>
> **Q2: Instead of the proposed Identity-Preserving Cross-Attention technique, is it possible to improve the consistency by performing SVR on C_P0, C_EOT**
> Your intuition is absolutely correct, and this approach does show potential. In our experiments, we explored concatenating C_{P0} and C_{EOT}, followed by applying SVR+ to enhance identity consistency (ID). While this method helped preserve ID, we observed that it could inadvertently impact the visual quality of the images.
>
> As an example in Fig.12 of the updated submission, performing SVR over P0 and EOT leads to high similarity of the backgrounds and poses across all frame images. In comparison, our method 1Prompt1Story shows better diversity in consistent T2I generation. We also aim to develop this idea in the future work.
>
> **Q3:  Does the proposed method affect the visual quality/image realism? How is it compared with other methods?**
> We evaluate the Fréchet Inception Distance (FID) metric to assess the quality of image generation. We use images generated by the base model as the real dataset and images generated by each method itself as the fake dataset. The results indicate that our 1Prompt1Story approach does not compromise the original model's image generation quality. Additionally, our method preserves the diversity of text-to-image (T2I) generation, as demonstrated in Appendix C.4 (Fig. 14), where we present results using various random seeds.
>
> | method | SD1.5  | SDXL | BLIP-Diffusion | Textual Inversion | The Chosen One | PhotoMaker | IP-Adapter | Consistory | StoryDiffusion | NPR | 1Prompt1Story (Ours) |
> | :--------: | :--------: |:--------: |:--------: |:--------: | :--------: | :--------: |:--------: |:--------: |:--------: |:--------: |:--------:
> | FID↓ | - |  - | 65.32 |  48.94 | 83.74 | 55.27 | 66.76 | 45.20 | 51.63 | 44.02 | 44.16 |

---

> > ### Comment · Reviewer_uG1g · 2024-11-26
> >
> > FID is not a metric specialized for visual quality and makes it difficult to compare between your method and the base model. Could you evaluate the aesthetic quality (using e.g., aesthetic score) and compare the base model, the baselines, and your method?

---

> > > ### Author Response · Authors · 2024-11-28
> > > **Thank you**
> > >
> > > Dear Reviewer uG1g:
> > >
> > > Thank you for your valuable feedback and suggestions. In response, we have conducted aesthetic score evaluations for the base model and all methods using the open-source code from https://github.com/christophschuhmann/improved-aesthetic-predictor. This tool, an enhanced version of the LAION aesthetic predictor [A, B], has been widely used for evaluating T2I diffusion models.
> > > Additionally, we have incorporated the ImageReward metric [C] to further assess human aesthetic preference and prompt alignment. These results indicate that our 1Prompt1Story does not compromise the base model's image generation quality.
> > >
> > > Once again, thank you for your time and thoughtful response.
> > >
> > > | method | SD1.5  | SDXL | BLIP-Diffusion | Textual Inversion | The Chosen One | PhotoMaker | IP-Adapter | Consistory | StoryDiffusion | NPR | 1Prompt1Story (Ours) |
> > > | :--------: | :--------: |:--------: |:--------: |:--------: | :--------: | :--------: |:--------: |:--------: |:--------: |:--------: |:--------:
> > > | aesthetic score↑ | 5.484 |  6.190 | 5.520 |  6.109 | 6.067 | 5.932 | 6.110 | 6.115 | 6.045 | 6.148 | **6.176** |
> > > | ImageReward↑ | 0.11 |  1.15 | -0.71 | 0.57 | 0.19 | 0.86 | 0.52 | **1.14** | 0.96 | 1.00 | **1.14** |
> > >
> > >
> > > [A] LAION. Laion Aesthetic Score Predictor. https://laion.ai/blog/laion-aesthetics/, 2023.
> > > [B] Laion-5B: An Open Large-Scale Dataset for Training Next Generation Image-Text Models.
> > > [C] ImageReward: Learning and Evaluating Human Preferences for Text-to-Image Generation. Jiazheng Xu, Xiao Liu, Yuchen Wu, Yuxuan Tong, Qinkai Li, Ming Ding, Jie Tang, Yuxiao Dong (NeurIPS 2023).
> > >
> > > Best Regards,
> > > Authors of submission 903

---

> > > > ### Comment · Reviewer_uG1g · 2024-11-29
> > > >
> > > > Thank you for the new results. I would like to maintain my rating and recommend accepting the submission.

---

> > > > > ### Author Response · Authors · 2024-11-29
> > > > > **Thank you**
> > > > >
> > > > > Dear Reviewer uG1g,
> > > > >
> > > > > We sincerely thank you for your kind and encouraging feedback. We truly appreciate the time and effort you dedicated to reviewing our work, and we're pleased that our response addressed your concerns. Your recognition of our efforts serves as great motivation for us to continue improving and advancing our research.
> > > > >
> > > > > Once again, thank you for your valuable time and support.
> > > > >
> > > > > Best Regards,
> > > > > Authors of submission 903

---

### Official Review · Reviewer_Q3y4 · 2024-11-04

**Soundness:** 3
**Presentation:** 3
**Contribution:** 2
**Rating:** 8
**Confidence:** 4

**Summary:**

The goal of this article is pretty similar to Cosistory, but it proposed a new training-free method which focus on how to leverage prompts better. This article first proposes Naive Prompt Reweighting which rescaling the embeddings of the other frame prompts by a reduction factor. Then, it proposes  Singular-ValueReweighting+&- which perform a SVD decomposition and Identity-PreservingCross-Attention(IPCA) which enhance the identity similarity between images generated from the concatenated prompt.

Work is relatively well done, simple and effective, but not very innovative(Reweight and Identity-Preserving or their similar ideas have appeared in previous articles, this artivle just use them into the enhancements of prompts)

What's more, I wonder the quality and improvements for multi-IDs, because nowadays the consistency works  can  generally do well with multiple IDs.

**Strengths:**

1. Drawing, good.
2.Focus on the enhancements of prompts and make it effectiveinstead of focus on images.
3.Proposed different reweightmethods： NPR and SVR+-, make the article more solid.

**Weaknesses:**

1. Reweight and Identity-Preserving or their similar ideas have appeared in previous articles, this artivle just use them into the enhancements of prompts(I remember Reweight is used in video, the extent how a prompt or ref img affects different frames is reweighted. )(Identity-Preserving--like [1] which operate on the modality of images)
2. The improvements for multi-IDs are NOT mentioned, which is always talked in recent works like StoryDiffusion(The example of a man and a dog).



[1] JeDi: Joint-Image Diffusion Models for Finetuning-Free Personalized Text-to-Image Generation(CVPR24)

**Questions:**

1. Prove your novelty (I know it's a little hard for training-free methods in this field)。
2. Show results on milti-IDs.

---

> ### Author Response · Authors · 2024-11-20
> **Response to Reviewer Q3y4**
>
> Thank you for your detailed review and for offering such thoughtful and constructive feedback. Below, we present a point-by-point response to the weaknesses and questions you mentioned.
>
> **W1: Reweight and Identity-Preserving or their similar ideas have appeared in previous articles**
> In this paper, we include Naive Prompt Reweighting (NPR) as a baseline method. However, NPR has proven ineffective in story generation scenarios. Our proposed SVR method represents a stronger reweighting approach. Additionally, while common prompt weighting methods operate at the single-word level, our NPR baseline uniquely functions at the sentence level. We are the first to propose an enhanced reweighting technique like SVR, which we plan to extend to future applications.
>
> Regarding identity preservation, our 1Prompt1Story method leverages the inherent properties of text embedding spaces, distinguishing it from JeDi in several ways:
> **(1) Training-free approach**: Our method is inherently training-free as it relies on the identity information embedded within a single joint prompt. In contrast, JeDi depends on the creation of subject-grouped datasets for training coupled self-attention layers.
> **(2) Objective focus**: While 1Prompt1Story ensures consistent T2I generation, JeDi targets T2I personalization. These methods are orthogonal and can complement each other. For instance, our method combined with PhotoMaker (as shown in Fig.8-right) can improve identity consistency while enabling real-image personalization.
> **(3) Generalizability**: 1Prompt1Story is based on the foundational properties of language models, making it applicable to a broader range of scenarios. Conversely, JeDi relies on an LLM-augmented pipeline to generate large-scale, grouped subject datasets.
>
> We already include JeDi in our related work and will incorporate this discussion to clarify these distinctions in the future version.
>
>
> **W2&Q2: The improvements for multi-IDs are NOT mentioned, which is always talked in recent works like StoryDiffusion(The example of a man and a dog).**
> By specifying multiple characters in the identity prompt, our method naturally extends to achieve multi-character ID consistency within a single generation. For instance, as demonstrated in Appendix D.1 (Fig. 15), we showcase results generated by our approach featuring two distinct characters, further validating its capability to handle complex identity setups.
>
> **Q1: Prove your novelty (I know it's a little hard for training-free methods in this field)**
> (1) We innovatively **utilize a single-prompt approach instead of the conventional multi-prompt setups** to achieve identity-consistent story generation. Unlike prior methods, which overlooked the inherent context consistency property of language models, our work is the first to explore and leverage this characteristic (Section 3.1).
> (2) Both our Singular Value Reweighting (SVR) and Identity-Preserving Cross-Attention (IPCA) techniques are **rooted in the context consistency observation**. SVR refines the expression of the current frame’s prompt while attenuating contributions from other frames, ensuring frame-specific focus. Simultaneously, IPCA strengthens subject consistency within the cross-attention layers, preserving identity across frames. By applying these techniques, **1Prompt1Story** achieves significantly more consistent T2I generation results compared to existing approaches.

---

> > ### Author Response · Authors · 2024-12-02
> > **Genuinely looking forward to your reply**
> >
> > Dear Reviewer Q3y4,
> >
> > Thank you for dedicating your time to review our paper and your valuable feedback. We are pleased to address all your concerns in the rebuttal period. Given the extension of the discussion period, we would greatly appreciate any further feedback, so we can strengthen the paper and allow the committee to make a more confident decision.
> >
> > Best Regards,
> > Authors of submission 903

---

### Official Review · Reviewer_Pz4Z · 2024-11-05

**Soundness:** 3
**Presentation:** 3
**Contribution:** 2
**Rating:** 5
**Confidence:** 4

**Summary:**

The paper proposes a training-free method to enhance subject consistency in text-to-image models. This setting comprises an identity prompt describing the desired subject and a set of frame prompts describing the subject in different scenarios. The proposed method consists in concatenating the identity prompt and all frame prompts into a single, longer, consolidated prompt. Then, in order to generate images for each frame prompt, the consolidated prompt embeddings are reweighed accordingly. This improves identity preservation but harms text-image alignment, so authors propose a couple of techniques to mitigate it. First, the consolidated prompt embeddings are reweighed based on their singular values, which helps preventing information mixing from different frames. Second, the cross-attention between text embeddings and latent image features is modified to improve the subject's consistency. The proposed approach is evaluated on a proposed benchmark and compared against several existing methods, outperforming other training-free methods for identity preservation.

**Strengths:**

* The proposed method is training free and only increases inference-time compute moderately.
* The proposed method outperforms the considered training-free existing methods on prompt alignment and identity preservation.
* The ablation study confirms the effectiveness of singular-value reweighing and identity-preserving cross-attention.

**Weaknesses:**

* Stable diffusion models use a CLIP text encoder, which is limited to 77 tokens. This severely constrains the number of frame prompts that can be used.
* Evaluating on a single benchmark of 200 examples seems insufficient. Additional benchmarks would be needed to verify the robustness of the results. Similarly, a human study on just 20 examples seems quite limited. I would suggest applying bootstrapping to obtain the sample variance.
* In figure 8 (right), applying the proposed method doesn't seem to have a big impact in terms of identity preservation.

**Questions:**

* In figure 2, why doesn't P1 have the same text embedding for multi-prompt and single-prompt settings? I believe it should since the context is the same (`[[SOT]; P0]`) and CLIP's text encoder has causal attention.
* What do the shaded regions in figure 2 represent? How are they computed?
* Does changing the order of frame prompts affect results? For instance, always placing the frame prompt of interest at the end.
* In figure 3, I don't see the NPR results as incorrect since the background isn't specified in frame prompts 2, 3, 4 and 5.
* Why does the embedding of the [EOT] token belong to both the expressed and suppressed sets? Won't the effects cancel each other out?
* For prompt alignment evaluation, I would suggest using more recent metrics that are more correlated with human judgment, such as DSG [1] or VQAScore [2].

[1] Cho, Jaemin, et al. "Davidsonian Scene Graph: Improving Reliability in Fine-grained Evaluation for Text-to-Image Generation." The Twelfth International Conference on Learning Representations.
[2] Lin, Zhiqiu, et al. "Evaluating text-to-visual generation with image-to-text generation." European Conference on Computer Vision. Springer, Cham, 2025.

---

> ### Author Response · Authors · 2024-11-20
> **Response to Reviewer Pz4Z -- Part1**
>
> Thank you for your thorough review and for providing such valuable and insightful feedback. Below, we provide a point-by-point response to the weaknesses and questions you raised.
>
> **W1: Stable diffusion models use a CLIP text encoder, which is limited to 77 tokens. This severely constrains the number of frame prompts that can be used.**
> This is also a key challenge addressed in our method. To counter the token length limit of the CLIP model, we introduced a sliding window technique to overcome this limitation, enabling infinite-length story generation. The details of this technique are elaborated in Appendix D.2 and illustrated in Fig. 17.
>
> **W2: Evaluating on a single benchmark of 200 examples seems insufficient. Similarly, a human study on just 20 examples seems quite limited.**
> We agree with your observation and have taken steps to address this limitation. To overcome this drawback, we developed the **Consistory+ benchmark**, which is three times larger than the original benchmark [1] in terms of the number of examples, categories, and overall diversity. Additionally, we expanded our experiments by incorporating more examples using bootstrapping, as per your suggestion.
>
> For comparison, we selected the two best-performing methods and evaluated them under varying conditions. Specifically, we generated additional image sets using different random seeds, referred to as **sample2** and **sample3**, while retaining **sample1** as the original results from Table 1. Combined, these sets account for a total of **4,500 images (1,500 per sample set)**, providing a robust foundation for our analysis.
>
> | Metric    | ip_adapter (samples1) | ip_adapter (samples2) | ip_adapter (samples3) | storydiffusion (samples1) | storydiffusion (samples2) | storydiffusion (samples3) | 1prompt1story (samples1) | 1prompt1story (samples2) | 1prompt1story (samples3) |
> |-----------|------------------------|------------------------|------------------------|----------------------------|----------------------------|----------------------------|--------------------------|--------------------------|--------------------------|
> | **clip_t↑** | 0.8458                 | 0.8459                 | 0.8474                 | 0.8877                     | 0.8873                     | 0.8865                     | 0.8942                   | 0.8879                   | 0.8903                   |
> | **clip_i↑** | 0.9429                 | 0.9431                 | 0.9415                 | 0.8755                     | 0.8816                     | 0.8858                     | 0.9117                   | 0.9142                   | 0.9118                   |
> | **dream sim↓** | 0.1462               | 0.1477                 | 0.1532                 | 0.3212                     | 0.3108                     | 0.3038                     | 0.1993                   | 0.1943                   | 0.1999                   |
>
> For the initial user study, we randomly selected 20 prompt sets across various categories, with each example comprising story generation images from four methods. The detailed information about the user study is introduced in Appendix.F and illustrated in Fig. 19.
>
> To enhance the reliability and breadth of the study, **we expanded it to include 37 questionnaires, each with 30 prompt sets of questions**. The updated results, summarized below, provide more robust evidence of the effectiveness of our approach across diverse scenarios.
>
> | method | 1Prompt1Story (Ours)  | StoryDiffusion | Consistory | IP-Adapter |
> | :--------: | :--------: |:--------: |:--------: |:--------: |
> | User Preference (%) | 48.6 |  29.8 |  13.0  |  8.6  |
>
> [1]Yoad Tewel, Omri Kaduri, Rinon Gal, Yoni Kasten, Lior Wolf, Gal Chechik, and Yuval Atzmon. 2024. Training-Free Consistent Text-to-Image Generation. ACM Trans. Graph. 43, 4, Article 52 (July 2024), 18 pages. https://doi.org/10.1145/3658157

---

> > ### Author Response · Authors · 2024-11-20
> > **Response to Reviewer Pz4Z -- Part2**
> >
> > **W3: In figure 8 (right), applying the proposed method doesn't seem to have a big impact in terms of identity preservation.**
> > First of all, PhotoMaker[1] focuses primarily on maintaining identity consistency of human faces in T2I generation and has already demonstrated excellent performance. In Fig. 8, we illustrate that our method, 1Prompt1Story, can be seamlessly integrated with other methods, such as PhotoMaker, as an orthogonal enhancement. By calculating the identity consistency metrics over Fig. 8, we show that our approach not only preserves but also improves PhotoMaker's identity consistency under the storytelling scenarios. Our 1Prompt1Story enhances the narrative flow, resulting in more harmonious and cohesive identity generation throughout the story. **Additional image examples are provided in the updated Appendix (Fig. 16).**
> >
> > | Metric                  | clip image↑ | dreamsim↓ |
> > |-------------------------|------------|----------|
> > | **PhotoMaker w/o ours (fig 8)**  | 0.7028     | 0.5530   |
> > | **PhotoMaker w/ ours (fig 8)**   | 0.8119     | 0.3641   |
> >
> >
> > [1]Li, Zhen, Cao, Mingdeng, Wang, Xintao, Qi, Zhongang, Cheng, Ming-Ming, and Shan, Ying. "PhotoMaker: Customizing Realistic Human Photos via Stacked ID Embedding." Proceedings of the IEEE/CVF Conference on Computer Vision and Pattern Recognition (CVPR), June 2024, pp. 8640-8650.
> >
> >
> > **Q1: In figure 2, why doesn't P1 have the same text embedding for multi-prompt and single-prompt settings?**
> > Yes, you are correct. When directly comparing the text embeddings, the value of P1 in both settings would indeed be identical and should correspond to the same point in a PCA visualization. However, t-SNE is a non-linear dimensionality reduction method and relies on the overall context during decomposition, which makes it suitable to measure the internal prompts similarities. This means it cannot guarantee that two identical points in the high-dimensional space will occupy the same position after the transformation.
> >
> > **Q2: What do the shaded regions in figure 2 represent? How are they computed?**
> > Apologies for the confusion caused by our earlier depiction. The shaded regions in the diagrams were manually drawn to better illustrate the distribution of points under the two settings; they were not indicative of any precise numerical data or statistical significance. In the revised submission Fig.2, we have replaced these shaded regions with direct lines to more accurately represent the intended relationships and eliminate any potential misunderstandings.
> >
> > **Q3: Does changing the order of frame prompts affect results? For instance, always placing the frame prompt of interest at the end.**
> > Thank you for pointing out this concern—it’s indeed a valid consideration when working with prompt order sensitivity. To address this issue, we included an ablation study presented in **Appendix C.1 (Fig. 10)** of the original submission. For this analysis, we tested all six possible order combinations for an example with three frame prompts. The visualization results indicate that the order of frame prompts has minimal impact on the generated output images. This consistency across permutations demonstrates the robustness of 1Prompt1Story to variations in frame prompt order.

---

> ### Author Response · Authors · 2024-11-20
> **Response to Reviewer Pz4Z -- Part3**
>
> **Q4: In figure 3, I don't see the NPR results as incorrect since the background isn't specified in frame prompts 2, 3, 4 and 5.**
> Thank you for bringing up this point. We agree that the lack of explicit background specification can lead to inconsistent or suboptimal generation, particularly with the NPR method. As noted, the 2nd and 4th frames in Fig.3 indeed demonstrate worse identity coherence when using NPR, reinforcing the need for a more robust handling of such scenarios. Our preset aligns with the original SDXL's behavior: in cases where no background is specified, it is preferable not to generate one. This ensures better focus on the primary content of the image. However, **as highlighted in Fig. 13 and Fig. 20 (the second-to-last row), when backgrounds are explicitly specified in the frame prompts, NPR still encounters background confusion problems**, compromising both spatial and thematic consistency across frames. This underscores the limitations of NPR when dealing with multi-frame prompts, particularly in complex storytelling scenarios. Our method 1Prompt1Story addresses this issue by prioritizing both identity consistency and background coherence, as demonstrated through our proposed approach in contrast with NPR in these figures.
>
> **Q5: Why does the embedding of the [EOT] token belong to both the expressed and suppressed sets? Won't the effects cancel each other out?**
> The EOT token's capacity to encapsulate all preceding text aligns with observations in prior research [1,2]. By aggregating the cumulative frame prompt information, the EOT token serves as a dependable anchor for achieving consistent T2I generation. Our SVR method utilizes this property effectively, isolating and amplifying the expressed components while diminishing the suppressed ones, all while preserving coherence across frames.
>
> These two mechanisms do not cancel each other out during the enhancement. The enhanced prompt is combined with the EOT token, followed by singular value decomposition (SVD), where only the main singular values are selectively amplified. This approach assumes that the main singular values correspond to the essential components of the frame prompt requiring enhancement. Similarly, suppression applies the same principle but focuses on weakening less relevant components. Ablation experiments conducted with **SVR+** and **SVR-** independently (Fig. 7, Table 3) demonstrate that combining both operations achieves superior results, highlighting the synergy between the two mechanisms.
>
> [1]Senmao Li, Joost van de Weijer, Fahad Khan, Qibin Hou, Yaxing Wang, et al. Get what you want, not what you don’t: Image content suppression for text-to-image diffusion models. (ICLR2024)
> [2]Yinwei Wu, Xingyi Yang, and Xinchao Wang. Relation rectification in diffusion model. (CVPR 2024)
>
> **Q6: For prompt alignment evaluation, I would suggest using more recent metrics that are more correlated with human judgment, such as DSG [1] or VQAScore [2].**
> Thank you for your suggestion. In our method, we have already incorporated the commonly used DreamSim metric to capture human preferences and identity consistency, and the CLIP-T metric is to measure the prompt alignment. Following your advice, we conducted additional comparisons of our approach with other methods using the DSG[1] and VQAScore[2] metrics. The results are presented below, where the SDXL model is listed as the baseline reference not for comparison. These evaluation results, along with the experiments in Table 1, prove the effectiveness of our method 1Prompt1Story in consistent T2I generation.
> | Metric                          | SD1.5  | SDXL   | BLIP-Diffusion | Textual Inversion | The Chosen One | PhotoMaker | IP-Adapter | ConsiStory | StoryDiffusion | NPR    | Ours   |
> |---------------------------------|--------|--------|----------------|-------------------|----------------|------------|------------|------------|----------------|--------|--------|
> | **VQAScore** ↑                 | 0.7157 | 0.8473 | 0.5735         | 0.6655            | 0.6990         | 0.8178     | 0.7834     | 0.8184     | **0.8335**     | 0.8044 | 0.8275 |
> | **DSG w/ dependency** ↑        | 0.7354 | 0.8524 | 0.6128         | 0.7219            | 0.6667         | 0.8108     | 0.7564     | 0.8196     | 0.8400         | 0.8407 | **0.8520** |
> | **DSG w/o dependency** ↑       | 0.8095 | 0.8961 | 0.6909         | 0.8051            | 0.7495         | 0.8700     | 0.8122     | 0.8696     | 0.8853         | 0.8863 | **0.8945** |
>
> [1] Cho, Jaemin, et al. "Davidsonian Scene Graph: Improving Reliability in Fine-grained Evaluation for Text-to-Image Generation." ICLR 2024.
> [2] Lin, Zhiqiu, et al. "Evaluating text-to-visual generation with image-to-text generation." ECCV 2024

---

> ### Comment · Reviewer_Pz4Z · 2024-11-26
>
> Thank you for your thorough response. While some of my concerns have been addressed, I still have reservations regarding the significance of the results. In addition, the justification for expressing and suppressing the EOT token remains unclear. As such, I've increased my soundness score to 3, but I'm maintaining my overall rating at 5, as I'm still not convinced about the impact of this work.
>
> I would like to clarify that [bootstrapping](https://en.wikipedia.org/wiki/Bootstrapping_(statistics)) is a technique to estimate the variance of a sample, not to increase the sample size. In addition, providing the means of 3 samples is not enough to compute the variance of each method and assess whether the improvements are statistically significant. For instance, the prompt alignment values between storydiffusion and 1prompt1story are very similar for the 3 samples.
>
> In figure 8, I believe the identity preservation metrics are worse for the original PhotoMaker because in the second image the subject is not facing the camera. In addition, it seems that applying the proposed method reduces diversity (the subject has a very similar pose in images 1 and 3), which is not captured in the quantitative results.
>
> In figure 2, I believe you could solve the issue by transforming all embeddings from both methods at the same time. Then the two sets of projected embeddings would be directly comparable. In addition, the same issue with shaded regions is present in figure 3, I encourage authors to fix that one as well.
>
> For the prompt order ablation, testing on a single example is not convincing. To provide evidence that the method is robust to the frame prompts' order, it needs to be validated across a dataset of prompts by computing quantitative results.
>
> Finally, the fact that VQAScore assigns a higher prompt alignment score to storydiffusion is in line with my previous concern that 1prompt1story might not be significantly better than storydiffusion in terms of prompt alignment.

---

> > ### Author Response · Authors · 2024-11-28
> > **New Response to Reviewer Pz4Z -- Part1**
> >
> > Dear Reviewer Pz4Z:
> >
> > Thank you for your further feedback and suggestions. The questions you’ve raised have been invaluable in helping us improve our paper. Below, we will answer your questions and concerns paragraph by paragraph (using "P" as shorthand for "paragraph"). Thank you again for reviewing our responses.
> >
> > **P1:  the justification for expressing and suppressing the EOT token remains unclear.**
> >
> > The reason for updating the text embedding’s EOT is that EOT **encapsulates the semantic information of the entire prompt [1,2]**, due to the self-attention mechanism in CLIP. Below, we provide an example to explain the role of EOT, and why the suppression and expression processes do not cancel each other out. Suppose we have four frame prompts: A, B, C, and D. We want to express the semantics of A while suppressing the semantics of B, C, and D.
> >
> > In the SVR+ process, we begin by concatenating the text embedding of A with the EOT, forming M = [A, EOT]. We then perform Singular Value Decomposition (SVD) on M and enhance the dominant singular values of M, which primarily correspond to the semantics of frame prompt A(which is proved in [1]). This results in a new M' (M' = [A', EOT']), where the semantic information of A is amplified in both A' and EOT'. In the SVR- process, we iteratively suppress the semantics of B, C, and D. First, we concatenate the embedding of B with the updated EOT' to form M = [B, EOT'], and apply SVD to suppress the dominant singular values that mainly represent the semantics of B. This gives us a new M' (M' = [B', EOT'']), where the semantic information of B is weakened in both B' and EOT''. We repeat this process similarly for C and D.
> >
> > **For each SVD decomposition, we suppress or express different semantic parts of the EOT.** Through this approach, we could suppress the unwanted semantics in EOT while preserving the desired ones. Additionally, we have included a simple ablation study on EOT, as shown in Appendix Fig. 13. The results demonstrate that **without any modifications to EOT, the generated images tend to mix the semantics of other frame prompts**.
> >
> > [1]Get what you want, not what you don’t: Image content suppression for text-to-image diffusion models. Senmao Li, Joost van de Weijer, Fahad Khan, Qibin Hou, Yaxing Wang, et al.(ICLR2024)
> > [2]Relation rectification in diffusion model. Yinwei Wu, Xingyi Yang, and Xinchao Wang. (CVPR 2024)
> >
> >
> > **P2-1: I would like to clarify that bootstrapping is a technique to estimate the variance of a sample, not to increase the sample size.**
> >
> > Thank you for your detailed explanation of the bootstrapping technique. We then calculated the clip-T, clip-I, and Dreamsim metrics based on the Bootstrapping approach (using a sample size of 10000). The corresponding mean and variance values are shown in the table below. The mean values are consistent with the results in Table 1, and the variances for the three methods are relatively small. This indicates that our method shows stable improvements over the training-free methods on these three metrics.
> > | Method          | Clip-T             | Clip-I             | DreamSim           |
> > |-----------------|--------------------|--------------------|--------------------|
> > | IP-Adapter      | 0.8458±2.107e-5    | 0.9428±5.650e-5    | 0.1461±2.639e-5    |
> > | StoryDiffusion  | 0.8876±1.821e-5    | 0.8755±1.569e-5    | 0.3212±5.557e-5    |
> > | 1Prompt1Story   | 0.8942±2.017e-5    | 0.9116±1.348e-5    | 0.1993±6.127e-5    |
> >
> >
> > **P2-2: In addition, providing the means of 3 samples is not enough to compute the variance of each method and assess whether the improvements are statistically significant.**
> >
> > We acknowledge that the number of samples in our initial evaluation was indeed too small. To improve the robustness of our results, we have increased the samples size to 20, with each sample comprising 10 prompt sets randomly selected from the Consitory+ benchmark. The mean and variance are presented in the table below. Similarly, our method also shows a stable improvement over StoryDiffusion.
> >
> > | Method           | Clip-T        | Clip-I         | DreamSim       |
> > |------------------|----------------|----------------|----------------|
> > | IP-Adapter       | 0.8374±0.000511 | 0.9470±0.000066 | 0.1367±0.000511 |
> > | StoryDiffusion   | 0.8855±0.000166 | 0.8796±0.000216 | 0.3134±0.000853 |
> > | 1Prompt1Story    | 0.8904±0.000279 | 0.9123±0.000117 | 0.1895±0.000561 |

---

> > > ### Author Response · Authors · 2024-11-28
> > > **New Response to Reviewer Pz4Z -- Part2**
> > >
> > > **P3: In figure 8, I believe the identity preservation metrics are worse for the original PhotoMaker because in the second image the subject is not facing the camera. In addition, it seems that applying the proposed method reduces diversity.**
> > >
> > > To provide more reliable results, we have conducted tests on Photomaker and 1Prompt1Story combined with Photomaker using the Consistory+ benchmark, including ID similarity and diversity metrics. The results are presented in the table below. For evaluating the diversity, we include the Density and Coverage metrics using Studio-Gan[1]. These metrics estimate the fidelity and diversity of the generated images using the pre-trained Inception-V3 model. To compute them, we make SDXL as the baseline to generate 500 images using the prompt in Fig.8 and compare them to PhotoMaker w/o 1Prompt1Story and PhotoMaker w/ 1Prompt1Story. **The results show that 1Prompt1Story w/ PhotoMaker performs similarly to PhotoMaker on the diversity metrics, while providing a significant improvement in ID similarity compared to PhotoMaker.** As you advised, we also updated the image in the middle of Fig. 8 for Photomaker.
> > >
> > > | Method               | Clip-I ↑ | DreamSim ↓ | Density ↑ | Coverage ↑ |
> > > |-----------------------|--------------|------------|-----------|------------|
> > > | PhotoMaker  w/o 1Prompt1Story| 0.8465       | 0.3996     | 0.1293    | 0.230      |
> > > | PhotoMaker w/ 1Prompt1Story| 0.8955       | 0.2834     | 0.1172    | 0.258      |
> > >
> > > [1] https://github.com/POSTECH-CVLab/PyTorch-StudioGAN
> > >
> > > **P4: In figure 2, I believe you could solve the issue by transforming all embeddings from both methods at the same time. Then the two sets of projected embeddings would be directly comparable. In addition, the same issue with shaded regions is present in figure 3, I encourage authors to fix that one as well.**
> > >
> > > Thank you for your advice. We have revised Figure 3 by using direct lines to more clearly depict the intended relationships and prevent any possible misunderstandings. Additionally, we transformed all embeddings from both methods using T-SNE for visualization, as shown in Fig.18 in the Appendix, which indicates that single prompt setting exhibit more compact
> > > distribution than multi prompt. We also conducted a similiar statistical analysis on the benchmark (single prompt setting:101.42, multi prompt setting:121.50), which shows consistent conclusion as our previous analysis that frame prompts share more similar semantic information and identity consistency within the single-prompt setup.

---

> > > > ### Author Response · Authors · 2024-11-28
> > > > **New Response to Reviewer Pz4Z -- Part3**
> > > >
> > > > **P5: For the prompt order ablation, testing on a single example is not convincing. To provide evidence that the method is robust to the frame prompts' order, it needs to be validated across a dataset of prompts by computing quantitative results.**
> > > >
> > > > Thank you for your advice. We have tested the random frame prompt order on our Consistory+ benchmark. The results, as shown in the table below, indicate that the order of frame prompts has minimal impact on the prompt alignment and identity consistency metrics.
> > > >
> > > > | Method                              | Clip-T  | Clip-I  | DreamSim |
> > > > |-------------------------------------|---------|---------|----------|
> > > > | 1Prompt1Story                       | 0.8942  | 0.9117  | 0.1993   |
> > > > | 1Prompt1Story (random frame prompt order) | 0.8933  | 0.9141  | 0.1982   |
> > > >
> > > >
> > > > **P6: the fact that VQAScore assigns a higher prompt alignment score to storydiffusion is in line with my previous concern that 1prompt1story might not be significantly better than storydiffusion in terms of prompt alignment.**
> > > >
> > > > 1Prompt1Story achieves results very close to StoryDiffusion on the VQA metric (1Prompt1Story: 0.8275, StoryDiffusion: 0.8335). Beyond this metric, 1Prompt1Story surpasses StoryDiffusion in **CLIP-T**, **DSG**, **User preference**, and particularly in **CLIP-I** and **DreamSim**. In the context of storytelling, maintaining identity consistency is crucial, and our method's advantages in this area are evident in the **CLIP-I** and **DreamSim** metrics, where it consistently outperforms **StoryDiffusion**.
> > > >
> > > > Furthermore, while **StoryDiffusion** delivers impressive performance, it comes at the cost of significantly higher GPU memory usage and compromised image quality. As shown in **Table 1**, **Table 4**, and the **Table below**, our method achieves a more efficient trade-off between performance and resource consumption.
> > > >
> > > > In addition, we computed two additional metrics: the **aesthetic score** [1] and **ImageReward** [2]. These metrics assess image quality, human preference, and prompt alignment. The results demonstrate that **1Prompt1Story** consistently outperforms other methods, maintaining superior image generation quality across diverse aspects.
> > > >
> > > >
> > > > | Metric  | SD1.5  | SDXL | BLIP-Diffusion | Textual Inversion | The Chosen One | PhotoMaker | IP-Adapter | Consistory | StoryDiffusion | NPR | 1Prompt1Story (Ours) |
> > > > | :--------: | :--------: |:--------: |:--------: |:--------: | :--------: | :--------: |:--------: |:--------: |:--------: |:--------: |:--------:
> > > > | aesthetic score↑ | 5.484 |  6.190 | 5.520 |  6.109 | 6.067 | 5.932 | 6.110 | 6.115 | 6.045 | 6.148 | **6.176** |
> > > > | ImageReward↑ | 0.11 |  1.15 | -0.71 | 0.57 | 0.19 | 0.86 | 0.52 | **1.14** | 0.96 | 1.00 | **1.14** |
> > > >
> > > > [1]https://github.com/christophschuhmann/improved-aesthetic-predictor
> > > > [2]ImageReward: Learning and Evaluating Human Preferences for Text-to-Image Generation
> > > > Jiazheng Xu, Xiao Liu, Yuchen Wu, Yuxuan Tong, Qinkai Li, Ming Ding, Jie Tang, Yuxiao Dong (NeurIPS 2023)
> > > >
> > > > Thank you once again for your suggestions and feedback, which have been very helpful in improving our paper. We appreciate your time and feedback.
> > > >
> > > > Best Regards,
> > > > Authors of submission 903

---

> ### Author Response · Authors · 2024-12-02
> **Genuinely looking forward to your reply**
>
> Dear Reviewer Pz4Z,
>
> Thank you for dedicating your time to review our paper, and for your continuous engagement during the discussion period.
> Your constructive comments have been invaluable, and we genuinely hope to get feedback from you. Regarding the question you mentioned, we include the corresponding experiments and discussion as follows:
>
> 1. **Ablation Study on EOT**: Our experiments (Fig. 13 in the revised version) demonstrate that without modifying the EOT token, the generated images tend to mix semantics across different frame prompts.
>
> 2. **Bootstrapping**: We applied the bootstrapping method to calculate the **Clip-T**, **Clip-I**, and **DreamSim** metrics. The results are consistent with those in Table 1, showing stable improvements of our method over training-free approaches.
>
> 3. **PhotoMaker with and without 1Prompt1Story**: We tested **PhotoMaker** with and without **1Prompt1Story** using the Consistory+ benchmark, including ID similarity and diversity metrics. The results indicate that while diversity metrics are similar, **1Prompt1Story** significantly improves ID similarity compared to **PhotoMaker**.
>
> 4. **T-SNE Visualization**: We applied **T-SNE** to visualize all embeddings from both settings (Fig. 18 in the Appendix). This confirms our earlier analysis, showing that frame prompts exhibit more consistent semantic information and identity preservation in the single-prompt setup.
>
> 5. **Prompt Order Ablation**: We tested random frame prompt orders on the Consistory+ benchmark, and the results show that prompt order has minimal impact on prompt alignment and identity consistency metrics.
>
> 6. **Aesthetic Score and ImageReward Metric**: We computed two additional metrics, **aesthetic score** [1] and **ImageReward** [2], to evaluate image quality, human preference, and prompt alignment. The results demonstrate that **1Prompt1Story** outperforms other methods, consistently delivering higher image generation quality across various aspects.
>
> Thank you again for your time and effort in reviewing our work and looking forward to your response.
>
> Best Regards,
> Authors of submission 903

---

### Author Response · Authors · 2024-11-20
**General response to reviewers**

We appreciate all reviewers (**R1**=**Pz4Z**, **R2**=**Q3y4**, **R3**=**uG1g**, **R4**=**eHi5**, **R5**=**7cWp**) for their positive feedbacks.
They note that this paper is well-written (**R3, R4, R5**);
the training-free method is simple (**R2, R3**), novel (**R3, R4, R5**) and effective (**R1, R2, R3**);
that we present insightful and interesting analysis over the context consistency for prompt enhancement, (**R2, R4, R5**), therefore will be adapted to have broader implications (**R3, R4**);
that we provide a robust experimental evaluation to demonstrate that our method outperforms other approaches (**R1, R5**).
**We sincerely thank all the reviewers for their thoughtful and constructive feedback. The revised part in the resubmission paper is highlighted in *red* for your reference.**
Below we respond to general questions raised by reviewers.  We use **W** to abbreviate Weaknesses and **Q** to represent Questions.


**General Response 1: User study (R1-W2, R5-W3)**
For the initial user study, we randomly selected 20 prompt sets across various categories, with each example comprising story generation images from four methods. The detailed information about the user study is introduced in Appendix.F and illustrated in Fig. 19.

To enhance the reliability and breadth of the study, **we expanded it to include 37 questionnaires, each with 30 prompt sets of questions**. The updated results, summarized below, provide more robust evidence of the effectiveness of our approach across diverse scenarios.

| method | 1Prompt1Story (Ours)  | StoryDiffusion | Consistory | IP-Adapter |
| :--------: | :--------: |:--------: |:--------: |:--------: |
| User Preference (%) | 48.6 |  29.8 |  13.0  |  8.6  |


**General Response 2: Story generation with long prompts (R1-W1, R3-W1, R5-W1)**
This is also a key challenge addressed in our method. To counter the token length limit of the CLIP model, we introduced a sliding window technique to overcome this limitation, enabling infinite-length story generation. The details of this technique are elaborated in Appendix D.2 and illustrated in Fig. 17.

---

### Meta-Review · Area_Chair_MFPp · 2024-12-20

**Metareview:**

This paper has received ratings of 8, 8, 8, 8, 5 where the reviewers generally provided high scores, emphasizing the novelty and practical contributions of the proposed "One-Prompt-One-Story" framework. The reviews mentioned the paper’s technical rigor, clear presentation, and significant impact on consistent text-to-image (T2I) generation.

The authors introduces a novel training-free approach for consistent T2I generation, leveraging the inherent context consistency of language models. The proposed framework, 1Prompt1Story, consolidates all prompts into a single input and employs two innovative techniques: Singular-Value Reweighting (SVR) and Identity-Preserving Cross-Attention (IPCA). These techniques refine text-to-image alignment and maintain identity consistency across various frames.

Strengths:
- Novel Approach: The use of context consistency and training-free methodologies represents a significant advancement in T2I generation.
- The development of SVR and IPCA provides robust mechanisms to ensure identity preservation and text-image coherence.
- Comprehensive Evaluation: Extensive experiments, including qualitative, quantitative, and user studies, confirm the method’s strengths and effectiveness.
- Broad Applicability: The framework’s compatibility with various models and ability to generate multi-subject stories enhance its utility.

Weaknesses:
- Scalability Concerns: The method’s reliance on pre-defined prompts and constraints on input length might limit its scalability for very long narratives. However, the sliding window technique partially mitigates this issue.
- Complexity for End-Users: While effective, the technical sophistication of SVR and IPCA may present implementation challenges for practitioners without advanced expertise.
- Some ablation results (e.g., SVR order of operations) could benefit from more precise visual examples to elucidate the incremental advantages of each component.

**Additional Comments On Reviewer Discussion:**

Thanks to the authors and reviewers - the rebuttal period was handled constructively. The authors provided clear and detailed clarifications, especially regarding experimental configurations and theoretical motivations for their method. They addressed key reviewer concerns about scalability and clarified the technical nuances of SVR and IPCA. The inclusion of additional experimental results and qualitative comparisons strengthened the paper's claims and demonstrated responsiveness to feedback.

---

### Decision · Program_Chairs · 2025-01-22

Accept (Spotlight)